# SMT: Fine-Tuning Large Language Models with Sparse Matrices

**Haoze He**[*1], **Juncheng Billy Li**[*1,2], **Xuan Jiang**[3], **Heather Miller**[1,2]
[1]Carnegie Mellon University, [2]Two Sigma Investments, [3]University of California, Berkeley
{haozeh,junchenl,heather.miller}@cs.cmu.edu, {xuanjiang}@berkeley.edu

## Abstract

Various parameter-efficient fine-tuning (PEFT) methods, including LoRA and its variants, have gained popularity for reducing computational costs. However, there is often an accuracy gap between PEFT approaches and full fine-tuning (FT), and this discrepancy has not yet been systematically explored. In this work, we introduce a method for selecting sparse sub-matrices that aims to minimize the performance gap between PEFT vs. full fine-tuning (FT) while also reducing both fine-tuning computational costs and memory costs. We explored both gradient-based [1] and activation-based parameter selection methods to identify the most significant sub-matrices for downstream tasks, updating only these blocks during fine-tuning. In our experiments, we demonstrated that SMT consistently surpasses other PEFT baselines (e.g., LoRA and DoRA) in fine-tuning popular large language models such as LLaMA across a broad spectrum of tasks, while reducing the GPU memory footprint by 67% compared to FT. We also examine how the performance of LoRA and DoRA tends to plateau and decline as the number of trainable parameters increases, in contrast, our SMT method does not suffer from such issues.

## 1 Introduction

While the inherent generalization capability of Large Language Models (LLMs) is impressive, enhancing performance on downstream tasks often still necessitates fine-tuning (Ding et al., 2022; Chung et al., 2022). However, as the size of these LLMs increases, there is a pressing challenge to optimize the fine-tuning process for better computational efficiency and memory utilization. For example, fine-tuning a pre-trained LLaMA 7B model without CPU offloading[2] requires at least 58 GB of GPU vRAM—13.6 GB for trainable parameters, 40 GB for Adam optimizer states and gradients, and 2 GB for activations. This requirement makes fine-tuning on consumer-level GPUs such as the NVIDIA RTX 4090 with 24 GB of memory impractical (Zhao et al., 2024).

To address the prohibitive computational challenges of full parameter fine-tuning, many parameter-efficient fine-tuning (PEFT) methods have emerged over the past two years. LoRA and its variants (Hu et al., 2021; Zhao et al., 2024; Dettmers et al., 2024; Liu et al., 2024b;a; Su et al., 2023; Wang et al., 2024b;a) reparameterize the full model weight into low-dimensional proxy parameter through low-rank adaptation method and successfully reduce both the optimizer memory and computational costs. However, even in state-of-the-art (SoTA) PEFT research, results show a notable performance gap between low-rank reparameterization methods and full parameter tuning across many datasets (Liu et al., 2024a). Additionally, in this work, we report a less recognized phenomenon: low-rank adaptation methods tend to experience a performance plateau as the parameter count (rank r) increases.

On the other hand, previous studies have attempted to select a small subset of the parameters to fine tune through understanding the internal logic of LLMs. Some knowledge editing methods, such as Constrained fine-tuning (Zhu et al., 2020), ROME (Meng et al., 2022a), and MEMIT (Meng et al.,

---

[*]Equal Contribution.

[1]More details about gradient-based parameter selection methods are in Section 3 and Appendix B.

[2]Although some libraries such as Deepspeed can move the optimizer memory costs to CPU, it will also slow down the fine-tuning with extra I/O communication time (Rajbhandari et al., 2020; Aminabadi et al., 2022).

2022b), have shown that LLMs have memory sections located in distinct layers. These memories could be modified via fine-tuning (Zhu et al., 2020). These works observed that domain-specific knowledge can be distributed separately and sparsely among layers. Motivated by these observations, aiming to narrow the performance gap between PEFT and full fine-tuning, we proposed a *Sparse Matrix Tuning(SMT)* approach. We aim to fine-tune the most relevant sparse submatrices for optimal downstream performance. The gradient-based activation selection method is inspired by Fisher Information (Sung et al., 2021), and activation-based selection method is inspired by AWQ (Lin et al., 2024) which was previously applied for quantization. After comparing empirical performance of both GW-selection and AW-selection, we opted for gradient-based selection despite AW-selection could have saved us the full backward propagation in warm-up steps. (see Section §3 for details). Our findings differ from those of (Geva et al., 2020; 2022), as our experiments show that fine-tuning attention mechanisms, particularly the value vectors, is more efficient than fine-tuning other weight matrices, including MLPs, where (Geva et al., 2020; 2022) suggest memory is stored.

In our experiments, our Sparse Matrix Tuning (SMT) approach achieves better performance compared to LoRA and DoRA using same amount of trainable parameters. Additionally, SMT narrows the accuracy gap between full fine-tuning, overcomes the performance plateau of low-rank adaptation PEFT methods, and significantly outperforms LoRA and DoRA while utilizing less than 5% of trainable parameters. Our experimental results show that SMT consistently outperforms SoTA PEFT (including LoRA, DoRA, and SpIEL) methods by 2+ points when fine-tuning popular LLMs (e.g. LLaMA series base model[3]) on commonsense reasoning and arithmetic reasoning benchmarks. For layers without selected sub-matrices, SMT freezes these layers, saving all their backward propagation computational costs, parameters update computational costs, optimizer memory costs, and activation memory costs. For layers with selected sub-matrices, SMT reduces the computational costs of backward propagation and parameter updates, as well as the optimizer and activation memory costs, to less than 1% of those incurred by standard full tuning. Below are our additional contributions:

- **Language Model Anatomy:** We investigate the distinct impacts of attention mechanisms versus MLPs (Multi-Layer Perceptrons) in LLMs. Our findings indicate that fine-tuning the V vectors is the most effective among the Q, K, and V vectors in attention layers, and that fine-tuning attention layers is more crucial for downstream performance than MLP layers.

- **Large Language Model System Efficiency:** The SMT implementation significantly reduces the computational costs of backward propagation, parameter updates, optimizer memory, and activation memory during fine-tuning. Our implementation is open source [4].

## 2 BACKGROUND AND RELATED WORKS

Existing PEFT methods methods can be grouped into three main categories: addition, reparameterization and specification. The **Addition category**, which involves adding extra adapters (He et al., 2021), increases model size and slows down inference, so it falls outside the scope of this paper.

**Reparameterization category:** Many works on parameter-efficient fine-tuning (PEFT) (Mangrulkar et al., 2022) have aimed to improve efficiency and performance by only fine-tuning lower-dimensional adapters of model weights. Notable examples include LoRA (Hu et al., 2021), DoRA (Liu et al., 2024a), QLoRA (Dettmers et al., 2023), and several other variants (Liu et al., 2024b; Dettmers et al., 2023; Wang et al., 2024b;a). However, the results of these works still indicate a performance gap between PEFT methods and full fine-tuning (FT). Concurrent research (Biderman et al., 2024) empirically demonstrated that such a gap is difficult if not impossible to eliminate, they also notice the performance saturation issue of LoRA, as we will discuss in Section §4.2.

**Specification category:** Beside low-rank adaptation methods, sparsity-inspired approach is a natural alternative to reduce computational costs and memory footprint by finetuning only a subset of parameters. Previous works have also explored sparsity-inspired methods in transfer learning, model pruning, or fine-tuning, including Sparse Increment Fine-Tuning(SIFT) (Song et al., 2023), SHiRA (Bhardwaj et al., 2024), Fisher Mask (Sung et al., 2021), Random Masking (Xu & Zhang, 2024), Lottery-Ticket SFT (LT-SFT) (Ansell et al., 2021), SpIEL (Ansell et al., 2024), and Diff-Pruning (Guo et al., 2020). However, SIFT (Song et al., 2023), Fisher Mask (Sung et al., 2021), random masking Xu & Zhang (2024), and LT-SFT (Ansell et al., 2021) still require full backward propagation to compute all gradients, offering no speed advantage over full fine-tuning (FT). While

---

[3]Base model is not yet instruction tuned

[4]https://github.com/HectorHHZ/Sparse_Matrix_Tuning/

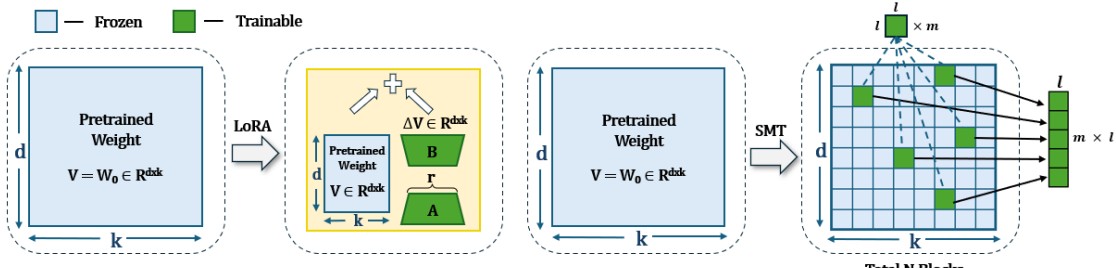

Figure 1: Differences between low-rank adaption method LoRA and SMT. The approach on the left illustrates the adaptation method used in LoRA, while the approach on the right represents the sub-matrix sparsity method utilized in SMT.

SpIEL (Ansell et al., 2024) partially reduces the need for backward propagation, its dynamic parameter selection still necessitates full backward propagation for a significant portion of training iterations. In LT-SFT (Ansell et al., 2021) and DiffPruning (Guo et al., 2020), the mask is applied at the layer level, without the option to selectively fine-tune individual parameters within the layer. Moreover, Sung et al. (2021); Ansell et al. (2021; 2024); Guo et al. (2020) require full fine-tuning phases and allocate both the model and the optimizer to GPUs, leading to substantial GPU memory consumption. The memory costs of Fisher Mask (Sung et al., 2021) are almost the same as the memory costs of the model parameters since the mask is stored as tensors with the same shape as the model parameters.(More details about Fisher Mask are in Appendix B.1). Furthermore, SIFT (Song et al., 2023) and SpIEL (Ansell et al., 2024) map discontinuous memory gradients to continuous memory addresses, creating a significant time bottleneck. Our work builds on existing strategies, but unlike previous methods, our matrix sparsity approach directly leverages task-specific gradient information to automatically adjust within-layer sparsity. This results in better speedup, reduced computational costs, and greater memory savings, as detailed in Section §3 and Section §4.4.

Suppose we are given a pre-trained auto-regressive language model $P_\Phi(y|x)$ parameterized by $\Phi$. Each downstream task is represented by a training dataset of context-target pairs: $Z = (x_i, y_i)_{i=1,...,N}$, where both $x_i$ and $y_i$ are sequences of tokens. Equation (1) dedicates the LoRA(Hu et al., 2021) fine-tuning process to maximize the conditional language modeling objective, which uses a low-rank representation to encode task-specific parameters. Specifically, LoRA freezes the pre-trained model weights and injects trainable rank decomposition matrices into each layer of the Transformer architecture. This is formulated as $\Delta\Phi = \Delta\Phi(\Theta)$, where $\Theta$ represents a much smaller-sized set of parameters with $|\Theta| \ll |\Phi_0|$. The resulting increment $\Delta\Phi$ can be as small as 0.01% of the pre-trained weights parameter size $|\Phi_0|$ in gradient updates. This greatly reduces the number of trainable parameters and the GPU memory requirement while maintaining model performance. [5]

$$\max_\Theta \sum_{(x,y)\in\mathcal{Z}} \sum_{t=1}^{|y|} \log(P_{\Phi_0+\Delta\Phi(\Theta)}(y_t|x, y_{<t})) \tag{1}$$

In our work, our proposed Sparse Matrix Tuning(SMT) uses matrix sparsity as the parameter-efficient approach. In SMT's case, reusing Equation( 1), the $\Theta$ represents the sub-matrices within the sparse weight matrices. SMT only fine-tunes sparse sub-matrices $\Theta$ instead of fine-tuning all pre-trained weights. Fig. 1 illustrates the differences between weight low-rank adaption method LoRA and our proposed sparse matrices tuning approach SMT. In SMT, we slice the pre-trained weight into $N$ sub-matrices and only fine-tune selected $M$ sub-matrices. The dimension of sub-matrices is $l \times l$, the total number of sub-matrices $N$ in a pre-trained weight is $N = \frac{d \times k}{l \times l}$. SMT constrains its update by representing the latter with a sparse gradient matrix $\Delta W_M$, $W_0 + \Delta W = W_0 + \Delta W_M$, where the number of fine-tuning sub-matrices $m \ll N$.

Since our proposed SMT method focuses on fine-tuning sub-matrices which are most relevant to downstream tasks' performance, identifying these sub-matrices is non-trivial. Our findings extend the observations from LoRA (Hu et al., 2021)—While low-rank adapters in LoRA are applied to the Q, K, V, and O vectors, the paper does not address the relative importance of components between MLP and attention mechanisms, nor does it determine which among Q, K, and V are the most critical during fine-tuning. In Section § 5, we explore these questions and provide experimental analysis to address them. Additionally, previous works (Zhu et al., 2020), MEMIT (Meng et al., 2022b), and

---

[5]More details about LoRA in Appendix F

(Geva et al., 2020; 2022) indicated that feed-forward MLP layers of the LLMs are most influential. However, through our experiment analysis§ 5.1, we found that fine-tuning attention layers is more efficient for improving downstream performance than MLP layers.

# 3 METHODOLOGY

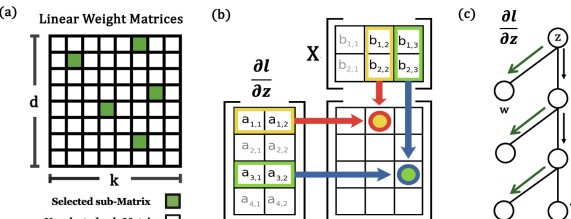

Figure 2: (a) A sparse weight matrix $W$. The green sub-matrices with significant gradients can be updated. (b) Backward propagation calculation for partial gradient for weight matrix $w$. (c) Computation graph in auto-differential systems.

Table 1: The experiments involved Full Fine-Tuning, SMT, LoRA, DoRA, and SpIEL on $4\times$ A100 40GB GPUs using data parallel, with a batch size of 16. Communication between the GPU and CPU was facilitated via PCIe-G4.

| LLaMA-7B | | | |
|---|---|---|---|
| PEFT method | #Params% | Time/s | Speedup |
| **Full Fine-tuning** | 100 | 243.84 | $1\times$ |
| SMT | 1.26 | 16.68 | $14.6\times$ |
| LoRA | 1.26 | 17.82 | $13.6\times$ |
| DoRA | 1.27 | 18.04 | $13.5\times$ |
| SpIEL | 1.26 | 25.45 | $9.6\times$ |

**Selection phase (warm-up phase):** The first step in our method is identifying the most salient sparse submatrices. We compare two strategies for this: Activation-aware selection (AW-selection) and Gradient-aware selection (GW-selection). Recent research (Sun et al., 2024) revealed the presence of massive activations in LLMs. Additionally, AWQ (Lin et al., 2024) found that using activation distribution, rather than weight distribution, more effectively identifies key weights, significantly reducing quantization errors by protecting these critical weights. A strong alternative is selecting sparse matrices based on Fisher Information (Sung et al., 2021) of the weight matrices. Building on empirical Fisher derivation (see Appendix B.1 for more details about Fisher information and how SMT differs with it), we propose GW-selection to identify specific submatrices within the model's weight matrices that show the greatest gradient changes during a warm-up phase (Fig. 2.a) at the beginning of fine-tuning. The GW-Selection warm-up phase lasts fewer than 100 iterations, varying depending on the dataset. Appendix B.2 details AW-Selection and Table 8 provides empirical evidence that it significantly underperforms GW-selection. Therefore, for the remainder of this paper, our SMT method selects sparse submatrices using the GW-selection approach identifying top z% submatrices (z=0.5)[6] with greatest gradient changes as critical submatrices.

**Efficient fine-tuning phase:** After identifying the submatrices for fine-tuning, SMT reduces memory requirements by freezing most of the weight matrices after the warm-up phase and storing the sparse weights in a compressed format. By implementing optimizer offloading, the warm-up phase of SMT doesn't cost extra GPU memory. After the warm-up stage, we disable offloading, move optimizers to GPUs, and use the `FusedAdam` optimizer from the `deepspeed` library (Aminabadi et al., 2022)to speed up fine-tuning.

**Advantages of SMT:** SMT significantly ***reduces backward computation costs*** to z% (z=0.5) of those in Full Fine-tuning (FT) by calculating gradients for only a subset of the weights during back-propagation. For linear layers in LLMs, where $Z = Wx$, the gradients with respect to weight matrix W and input x can be calculated as Equation (2):

$$\nabla_x f(x) = \frac{\partial l}{\partial Z} \cdot W; \qquad \nabla_W f(x) = \frac{\partial l}{\partial Z} \cdot x \tag{2}$$

where $\partial l/\partial z$ is the gradient information from backward propagation in (Fig. 2.b,c). $\nabla_w f(x)$ is the gradient matrix and and $x$ is the activation in the (Fig. 2.b). (Fig. 2.b) also illustrates that only partial backward computations are necessary when we update selected sparse matrices. To calculate the sub-matrix gradient (highlighted in yellow), it is only necessary to multiply the yellow row in $\partial l/\partial z$ with the yellow column in the activation $x$. Similarly, to calculate the green sub-matrix gradient, we only need to multiply the green row in $\partial l/\partial z$ with the green column in activation $x$. Note that in backward propagation, we only reduce computation when derivative to gradient matrix $w$ among as illustrated by the green arrows in (Fig. 2.c). but not other necessary computation. (black arrows)

---

[6]For LLaMA-7B and LLaMA2-7B, when the number of trainable parameter is 33685504, the percent of trainable parameter in whole model is 33685504/6738411520 $\approx$ 0.5%.

Besides, SMT reduces ***the memory costs of the optimizer gradients*** to z% (z=0.5). Since SMT only updates selected sparse sub-matrices, partial gradients are stored to cut the memory costs of the Adam optimizer to z% (z=0.5). This reduction is crucial because the memory cost of the Adam optimizer is typically twice the size of the model, which often consumes the majority of GPU RAM. SMT also reduces ***the gradient step computation costs*** to z% since only partial gradient steps are required.

Furthermore, SMT reduces ***the activation memory costs for the in forward pass*** to z% (z=0.5). Since SMT only computes the partial gradient, it saves the relevant portions of activations $X$ necessary for the gradient calculation as is represented in Equation. 2. In (Fig. 2.b), to calculate the green and yellow gradients in the gradient matrix, we only need to save the yellow and green columns of the activation $X$. It reduces the memory costs for the forward pass of the selected linear layer.

In SMT, all the layers except selected Q, K, and V vectors are ***frozen*** during fine-tuning. By doing this, SMT avoids all the weight backward propagation computational costs, parameters update computational costs, optimizer memory costs, and activation memory costs in frozen layers. The rationale for fine-tuning only the Q, K, and V vector is detailed in Section §5.1.

By applying sparse sub-matrix fine-tuning, SMT can reduce the fine-tuning memory costs of LLaMA-7B and LLaMA2-7B to less than 20GB and fit the fine-tuning into a 3090 24GB GPU. We also reduce the computation and achieve faster fine-tuning compared with FT and LoRA/DoRA, Section § 4.4 provides more details.

**Implementation:** In SMT, we first sum up gradients from the attention linear layers in every single warm-up iteration. The summed-up gradient information is used to identify task-specific sparse blocks. After the warm-up steps, we average the absolute values within the sub-matrices, select the sub-matrices with the largest value, and save the indices for the selected sub-matrices. In all of our experiments, we use $l \times l = 256 \times 256$ as sub-matrices block size. During the warm up steps, we can apply offload (Rajbhandari et al., 2020) on memory constraint GPU devices. Since SMT requires fewer than 100 warm-up steps in our experiments, it does not become a bottleneck during fine-tuning epochs. SMT updates the model during the warm-up phase to accelerate convergence. Additionally, SMT implements a custom sparse linear layer to ensure that unselected gradients are not calculated, saved, and updated (Code Snippet 6). We replace the selected linear layers with these customized sparse linear layers.

The custom sparse linear layer applies a specialized sparse linear multiplication function, integrated into our customized sparse linear layers (Code Snippet 7). This function calculates partial weight gradients based on the input, weight, and selected weight index. It significantly reduces the computational costs of backward propagation weight gradients to just z% (z=0.5) and minimizes the memory usage of the returned partial gradients to only z% (z=0.5).

The specialized sparse linear multiplication function rewrites both forward and backward functions. In the forward pass (Code Snippet 7) of sparse linear multiplication function, we only save selected activation $x$ using `ctx.save_for_backward()`, and in the backward pass (Code Snippet 8), we customize matrix multiplication to calculate the needed partial gradients given partial input and gradient index (shown in Fig. 2(b)). It is important to note that we do not use Sparse Matrix-Matrix Multiplication(SPMM)[7] because we concatenate the selected sparse sub-matrices and formed a $m \times l \times l$ dense matrix as illustrated in right part of Fig. 1. This would not costs additional time since memory allocations remain continuous within each sub-matrix. Despite employing matrix sparsity, we still leverage the advantages of dense matrix multiplication.

Furthermore, SMT gathers sparse matrix but still leverages dense matrix. SMT customizes the function for gathering trainable parameters. This customized function selectively gathers weight sub-matrices in the Q, K, and V vector layers and passes them to the Adam optimizer. By continuing to use the well-designed `FusedAdam` from the `deepspeed` library (Aminabadi et al., 2022), we maintain the computational speed of dense matrix weight updates. However, our approach reduces the gradient memory costs in the optimizer to just z% (z=0.5).

---

[7]Sparse Matrix-Matrix Multiplication is significantly slower than General Matrix Multiply. More details in Appendix G.

## 4 EXPERIMENTS AND RESULTS

**Model Architecture and Dataset:** In our experimental setup, we use open-weight LLaMA-7B, LLaMA-13B, LLaMA2-7B, and LLaMA3-8B models (AI@Meta, 2024). In Subsection§ 4.1§ 4.2, We perform fine-tuning on the Common Sense Reasoning tasks with 8 sub-tasks, each with a predefined training and testing set. We follow the setting of (Hu et al., 2023; Liu et al., 2024a) and amalgamate the training datasets from all 8 tasks to create the final training dataset `commonsense_170k` and conduct evaluations on the individual testing dataset for each task. We calculate an average score to encapsulate the overall efficacy. In Subsection§ 4.3, we perform fine-tuning on `Math10K` (Hu et al., 2023) dataset which includes `MultiArith`, `GSM_8K` (Cobbe et al., 2021), `AddSub`, `AQuA`, `SingleEq`, `SVAMP` datasets and evaluate the efficiency on their testsets.

**Training Framework and SMT Hyper-parameters:** We used the `DeepSpeed` (Aminabadi et al., 2022) library for fine-tuning and `accelerate` (Gugger et al., 2022) library for inference evaluation. Both training and fine-tuning are using `dtype` bf16. All experiments are fine-tuned for 3 epochs. In all our experiments in Section§ 4, sub-matrices are selected in blocks of size $l = 256$. We choose this specific sub-matrix dimension $l$ because it is the largest common factor of the column and row sizes of all linear layers in the LLaMA series models, using this dimension for slicing avoids remainder issues. We freeze all MLP layers and apply SMT only to the Q, K, and V vectors in the attention mechanism. In Section§5.1, we explain the rationale why we only apply SMT to attention mechanism instead of MLP. At the end of the gradient warm-up iteration, SMT ranks the average absolute gradient values within each sub-matrix and selects those with the highest average values. We determined the sub-matrix selection metric and gradient calculation method through ablation studies, incorporating metrics used in previous research (Sung et al., 2021). Ablation studies and the rationale of such selection are explained more in detail in Appendix B. We apply 100 warm-up iterations to all SMT experiments on `Commonsense` dataset and apply 25 warm-up iterations to all SMT experiments on `Math10K` dataset. The number of warm-up iterations is fine-tuned for each dataset. Detailed ablation studies on warm-up iterations are provided in Appendix H.

**PEFT Baselines:** For state-of-the-art (SOTA) baselines, we choose to include LoRA (Hu et al., 2021), DoRA (Liu et al., 2024a), and SpIEL (Ansell et al., 2024). LoRA and DoRA fall under the reparameterization category, while SpIEL, like SMT, belongs to the specification category. For hyper-parameter settings of LoRA, we follow the instructions suggested by (Biderman et al., 2024; Han & Michael; Kalajdzievski, 2023; Shih-yang). The LoRA adapters apply to $W_q, W_k, W_v, W_o, W_{gate}, W_{up}$, and $W_{down}$. $\alpha$ is determined by 2× rank.

**Computational Resources:** We conduct our experiments and implement SOTA baselines of LoRA (Microsoft) and DoRA (Shih-yang) to fine-tune LLaMA-7B and LLaMA2-7B model with 4 NVIDIA A100_40GB GPUs and fine-tune LLaMA-13B and LLaMA3-8B model with 4 NVIDIA A100_80GB GPUs. Communication between the CPU and GPU is facilitated via PCIe-G4 and communication between GPUs is facilitated via Nvlink-3.

**Evaluation Metrics:** We evaluate the performance of SMT in terms of computational efficiency (wall-clock time speedup), memory usage (analysis for memory complexity) in Subsection§ 4.4. In this section, we mainly evaluate SMT in terms of popular NLP tasks to test its ability to generalize to all downstream tasks. In Subsection§ 4.1§ 4.2, we evaluate the performance of SMT on 8 tasks in the `Commonsense` dataset, including `BoolQ`, `PIQA`, `SIQA`, `HellaSwag`, `ARC-e`, `ARC-c`, and `OBQA`, and we calculate an average score to encapsulate the overall efficacy. In Subsection§ 4.3, we perform fine-tuning on `Math10K` (Hu et al., 2023) dataset which includes `MultiArith`, `GSM_8K`, `AddSub`, `AQuA`, `SingleEq`, `SVAMP` datasets and evaluate the efficiency of SMT on their testsets. Both datasets, `Commonsense` and `Math10K`, focus on the generalization ability of LLMs across different sub-tasks, ensuring that our results are robust. All of the experiments are evaluated using accuracy.

### 4.1 COMMONSENSE REASONING

We evaluate SMT against the state-of-the-art (SoTA) weight low-rank adapter method includes LoRA and DoRA. To ensure a fair comparison, we fine-tuned model with SMT following the LoRA and DoRA configuration. We ensure all the hyper-parameters including batch size, data type, learning rate, and sequence length are identical to what was reported in LoRA and DoRA (Hu et al.,

Table 2: Accuracy comparison of LLaMA 7B, LLaMA 13B, LLaMA2 7B, and LLaMA3 8B with various PEFT methods on eight commonsense reasoning datasets. Results of all the baseline methods on LLaMA 7B, LLaMA 13B, LLaMA2 7B, LLaMA3 8B are taken from (Liu et al., 2024a). Results of all SMT are obtained using the hyper-parameters described in (Liu et al., 2024a) under the same settings. Bold texts dedicate the performance of SMT under the same numbers of parameters where LoRA, DoRA, and SpIEL achieve the best performance. Blue texts dedicate the best performance of SMT. Please note that the performance of LoRA, DoRA, and SpIEL *under larger numbers of trainable parameters can be found in Table 3.*

| Model | PEFT method | #Params% | BoolQ | PIQA | SIQA | HellaSwag | WinoGrande | ARC-e | ARC-c | OBQA | AVG |
|---|---|---|---|---|---|---|---|---|---|---|---|
| ChatGPT(175B) | - | - | 73.1 | 85.4 | 68.5 | 78.5 | 66.1 | 89.8 | 79.9 | 74.8 | 77.0 |
| LLaMA-7B | LoRA(Best) | 0.83 | 67.5 | 80.8 | 78.2 | 83.4 | 80.4 | 78.0 | 62.6 | 79.1 | 76.3 |
| | DoRA(Best) | 0.84 | 69.7 | 83.4 | 78.6 | 87.2 | 81.0 | 81.9 | 66.2 | 79.2 | 78.4 |
| | SpIEL(Best) | 0.84 | 67.7 | 81.2 | 78.6 | 84.0 | 80.2 | 78.3 | 62.8 | 78.8 | 76.5 |
| | SMT | 0.84 | 68.7 | 81.7 | 78.3 | 91.6 | 78.8 | 84.1 | 68.7 | 77.4 | **78.7** |
| | SMT(Best) | 4.91 | 72.0 | 82.9 | 80.7 | 93.3 | 82.4 | 86.1 | 70.6 | 83.0 | 81.4 |
| | Full Fine-tuning | 100 | 69.9 | 84.2 | 78.9 | 92.3 | 83.3 | 86.6 | 72.8 | 83.4 | 81.4 |
| LLaMA-13B | LoRA(Best) | 0.67 | 72.1 | 83.5 | 80.5 | 90.5 | 83.7 | 82.8 | 68.3 | 82.4 | 80.5 |
| | DoRA(Best) | 0.68 | 72.4 | 84.9 | 81.5 | 92.4 | 84.2 | 84.2 | 69.6 | 82.8 | 81.5 |
| | SpIEL(Best) | 0.68 | 73.2 | 84.3 | 81.4 | 91.2 | 84.1 | 83.1 | 68.8 | 82.8 | 81.1 |
| | SMT | 0.68 | 71.1 | 84.4 | 81.7 | 93.7 | 83.2 | 86.7 | 73.7 | 85.2 | **82.4** |
| | SMT(Best) | 4.91 | 72.6 | 86.1 | 81.9 | 95.0 | 86.1 | 88.2 | 77.1 | 87.4 | 84.3 |
| LLaMA2-7B | LoRA(Best) | 0.83 | 69.8 | 79.9 | 79.5 | 83.6 | 82.6 | 79.8 | 64.7 | 81.0 | 77.6 |
| | DoRA(Best) | 0.42 | 72.0 | 83.1 | 79.9 | 89.1 | 83.0 | 84.5 | 71.0 | 81.2 | 80.5 |
| | SpIEL(Best) | 0.83 | 70.5 | 80.6 | 80.8 | 85.8 | 83.4 | 81.2 | 65.8 | 81.8 | 78.3 |
| | SMT | 0.84 | 72.0 | 83.8 | 80.8 | 93.3 | 82.8 | 86.7 | 74.0 | 81.0 | **81.8** |
| | SMT(Best) | 4.91 | 72.6 | 85.2 | 82.0 | 94.4 | 85.7 | 87.8 | 74.5 | 85.0 | 83.4 |
| | Full Fine-tuning | 100 | 72.8 | 83.4 | 78.7 | 92.7 | 85.5 | 86.2 | 74.7 | 83.4 | 82.2 |
| LLaMA3-8B | LoRA(Best) | 0.70 | 70.8 | 85.2 | 79.9 | 91.7 | 84.3 | 84.2 | 71.2 | 79.0 | 80.8 |
| | DoRA(Best) | 0.71 | 74.6 | 89.3 | 79.9 | 95.5 | 85.6 | 90.5 | 80.4 | 85.8 | 85.2 |
| | SpIEL(Best) | 0.70 | 72.1 | 83.6 | 80.0 | 91.8 | 85.4 | 91.2 | 76.8 | 80.8 | 82.7 |
| | SMT | 0.71 | 75.7 | 88.4 | 81.4 | 96.2 | 88.2 | 92.7 | 83.2 | 88.6 | **86.8** |
| | SMT(Best)) | 3.01 | 75.1 | 89.9 | 82.4 | 96.3 | 88.8 | 92.6 | 82.8 | 89.6 | 87.2 |

2021; Liu et al., 2024a). We re-implemented LoRA and DoRA and achieved their best performance reported in (Liu et al., 2024a).

Table 2 demonstrates that SMT consistently surpasses baseline methods across LLaMA-7B, LLaMA13B, LLaMA2-7B, and LLaMA3-8B. Notably, by overcome plateau phenomenon, SMT further enhances accuracy of DoRA by 3.0%, 2.8%, 2.9%, and 2% on LLaMA-7B, LLaMA-13B, LLaMA2-7B, and LLaMA3-8B respectively. Notably, LoRA and DoRA will not achieve better performance with larger trainable parameters and exhibit the plateau phenomenon. In Subsection§ 4.2, we report and demonstrate the plateau issue in LoRA and DoRA and demonstrate SMT overcomes this issue. Moreover, by fine-tuning less than 5% of all parameters, SMT achieves similar accuracy performance of full fine-tuning while speedup 14.6× (speedup details in Table 1) and save 99.5% of optimizer memory(memory bottleneck in fine-tuning, details discussed in Section§ 3).

SMT can also consistently surpass LoRA and DoRA under the same number of trainable parameters where LoRA and DoRA achieve the best results, SMT can surpass their performance and also outstrip ChatGPT-3.5-turbo[8]. For instance, SMT consistently surpasses DoRA on LLaMA2-7B, LLaMA3-8B, LLaMA-13B, and LLaMA-7B by 1.3%, 1.6%, 0.9%, and 0.3% respectively, under their best performance trainable parameter number.

## 4.2 PLATEAU IN WEIGHT LOW RANK ADAPTION METHODS

In Table 3, we scale up the model size and presents how the performance of LoRA, DoRA, and SpIEL will be under larger number of trainable parameters. The corresponding visualization is provided in Fig. 9 in Appendix E. We reimplement all the experiments of LoRA (Microsoft), DoRA (Shih-yang), and SpIEL (Ansell et al.) using their official repository and followed their recommendation of hyper-parameters to achieve best performance under every single trainable parameter size. We observe that for SpIEL, DoRA, and LoRA models, with some larger ranks, their performance slightly degrades. However, SMT continues improving its performance when we scale up the trainable parameter size. When we scale up the trainable parameter size to 4.91%, SMT

---

[8]Results of ChatGPT-3.5-turbo are reported in DoRA (Shih-yang)

significantly surpass DoRA by 3.8% and 4.9% on LLaMA-7B and LLaMA-2-7B fine-tuned models. We postulate that such plateau phenomenon of LoRA or DoRA is due to their lossy low-rank approximation of the full weight information (includes lots of noise), whereas our SMT focuses on most prominent submatrices (contains less noise) and remains full rank gradient updates for the selected portion, making SMT performs better. SpIEL selects a fixed ratio of parameters from every layer. In contrast, SMT selects the most significant parameters, allows a different ratio of trainable parameters in each layer, and allocates more trainable parameters to more critical layers.

Table 3: Accuracy comparison of LoRA, DoRA, and SMT under different scaling of trainable parameters on `commonsense` datasets. Given certain base model and PEFT method, we gradually increase the number of trainable parameters on each line from left to right. On each line, the best performing model has *.

| | Method | 0.43 | 0.84 | 1.26 | 2.50 | 3.73 | 4.91 |
|---|---|---|---|---|---|---|---|
| LLaMA-7B | LoRA | 70.9 | 76.3* | 76.4 | 75.0 | 75.3 | 74.7 |
| | SpIEL | 72.6 | 77.4 | 78.2* | 76.8 | 77.2 | 76.4 |
| | DoRA | 77.5 | 78.4* | 76.0 | 77.3 | 77.5 | 77.6 |
| | SMT | 77.3 | 78.6 | 79.2 | 80.2 | 80.8 | 81.4* |
| LLaMA2-7B | LoRA | 76.5 | 77.6 | 78.4* | 77.6 | 77.3 | 77.0 |
| | SpIEL | 77.4 | 77.9 | 79.2* | 78.2 | 78.3 | 78.8 |
| | DoRA | 80.5* | 79.7 | 78.8 | 77.6 | 76.8 | 78.5 |
| | SMT | 81.1 | 81.8 | 81.7 | 82.2 | 82.8 | 83.4* |

Table 4: Fine-tuned LLaMA-7B model performance on `Commonsense`. AVG dedicates the average test score of eight subsets among `Commonsense`. MLP% and Attention% presents the percentage of trainable parameters apply to MLPs and attention mechanisms respectively.

| Model | MLP% | Attention% | AVG |
|---|---|---|---|
| | 0.84 | 0 | 76.7 |
| SMT(0.84%) | 0.42 | 0.42 | 77.3 |
| LLaMA-7B | 0.21 | 0.63 | 77.8 |
| | 0 | 0.84 | 78.7 |

## 4.3 OTHER DATASET

To ensure our findings above are generalizable, we further examine the performance of SMT under arithmetic reasoning dataset, `Math10K` (Hu et al., 2023). `Math10K` dataset has six subsets including `GSM8k`, `SingleEq`, `SVAMP`, `MultiArith`, `AddSub`, and `AQuA`. More details about `Math10K` dataset can be found in Appendix D. To ensure a fair comparison, we follow the open source hyper-parameter instruction in (Hu et al., 2023) to achieve best performance for LoRA and Dora, and apply the same hyper-parameters to SMT while only fine-tune the learning rate. Table 5 reports the performance of LoRA, DoRA, and SMT on the `Math10K` dataset[9] We can observe that SMT surpasses the best achievable performance of LoRA and DoRA by 1.3% and 1.1% respectively using the same amount of trainable parameters. In addition, by scaling up the trainable model size to 1.26%, SMT achieves better performance and surpasses the best performance of LoRA and DoRA by 2.5% and 2.3% respectively.

Table 5: SMT, LoRA and DoRA reproduction, and experiment results on `Math10K` dataset.

| Model | PEFT method | #Params% | GSM8k | SingleEq | SVAMP | MultiArith | AddSub | AQuA | AVG |
|---|---|---|---|---|---|---|---|---|---|
| LLaMA-7B | LoRA(Best) | 0.86 | 35.4 | 83.2 | 52.1 | 92.8 | 83.4 | 18.6 | 60.9 |
| | DoRA(Best) | 0.86 | 35.2 | 83.7 | 51.8 | 92.8 | 82.8 | 20.2 | 61.1 |
| | SMT | 0.86 | 34.2 | 84.6 | 53.6 | 91.5 | 85.8 | 23.6 | 62.2 |
| | SMT(Best) | 1.26 | 35.6 | 85.3 | 54.8 | 93.4 | 86.8 | 24.2 | 63.4 |

## 4.4 MEMORY AND COMPUTATION SAVING: SMT VS. LOW-RANK ADAPTION METHODS

SMT is *more computational efficient* than weight low-rank adaption method when the number of trainable parameters are the same, weight low rank adaption methods need to maintain additional adapters, which require additional forward computation. For instance, since LoRA maintains adapters $A$ and $B$, and the forward propagation is:

$$h = W_0 x + \Delta W_x = W_0 x + BAx \tag{3}$$

where the term $BAx$ requires additional forward propagation calculation, which is cut off in SMT. Regarding memory costs, since SMT does not require additional low-rank adapters $A$ and $B$, SMT can achieve *lower memory costs* than LoRA and DoRA under the same amount of trainable parameters setting. We illustrate this in Fig. 1. Taking the popular LLaMA-13B model as an example,

---

[9]In accordance with the special announcement for `Math10K` (Hu et al.), we include training set evaluation results from the `MultiArith`, `AddSub`, and `SingleEq` datasets, as well as test set evaluation results from the `GSM8k`, `SVAMP`, and `AQuA` datasets. `AVG` denotes the average accuracy across all evaluations.

since the model size is approximately 25 GB, if we fine-tune 1% of parameters, SMT can potentially save 250MB GPU memory compared to LoRA and DoRA. In Table 1, we provide the fine-tuning time costs for SMT, Full Fine-tuning, LoRA, and DoRA. SMT achieves an 14.6× speedup compared to Full Fine-tuning and outperforms both LoRA and DoRA. Additionally, compared to SpIEL in the specification category, SMT is 1.5× faster due to its more efficient selection phase, more flexible parameter selection, and improved trainable parameter memory management. We conducted time profiling by averaging the fine-tuning time every 10 iterations over 1000 iterations, following a 500-iteration warm-up period. Full fine-tuning utilized offload settings to accommodate the LLaMA model, which employs the Adam optimizer, within 40GB GPUs. SMT offers greater computational and memory efficiency, though these are secondary benefits compared to its primary focus.

## 5 FURTHER DISCUSSION

### 5.1 ATTENTION VERSUS MLP

In order to study what components are more critical for LLM's downstream performance during fine-tuning, we conduct ablation studies that compares MLPs vs. attention layers by adjusting the ratio of their trainable parameters respectively. We apply SMT and fine-tune 0.86% of parameters on LLaMA-7B using `Commonsense` dataset. In Table 4, we present four experiments. In the first row, all trainable parameters are allocated to MLPs. In the second row, both MLPs and Q, K, V vectors from attention mechanisms receive 0.43% of trainable parameters. In the third row, 0.62% of trainable parameters are assigned to Q, K, V vectors from attention mechanisms and 0.21% to MLPs. In the fourth row, all trainable parameters are dedicated to Q, K, V vectors from attention mechanisms. To guarantee a fair comparison, all the other hyper-parameters and settings are the same among these experiments.

In these experiments, allocating Y% of trainable parameters to MLPs or attention mechanisms means ranking the average absolute gradient values of each sub-matrix within the MLPs or attention mechanisms and selecting those with the highest average values until the number of parameters reaches Y%. The results reveal a significant performance gap between the first and fourth rows. The more trainable parameters we allocate to attention mechanisms, the better the fine-tuned model performs. When all SMT trainable parameters are applied to attention mechanisms, the model outperforms the one where all parameters are allocated to MLPs by 2.0%. Our empirical findings challenge previous assumptions (Zhu et al., 2020; Meng et al., 2022a; Geva et al., 2020; 2022) that the memory sections of large language models are primarily located in feed-forward MLP layers.

### 5.2 V VECTOR VERSUS Q, K VECTOR

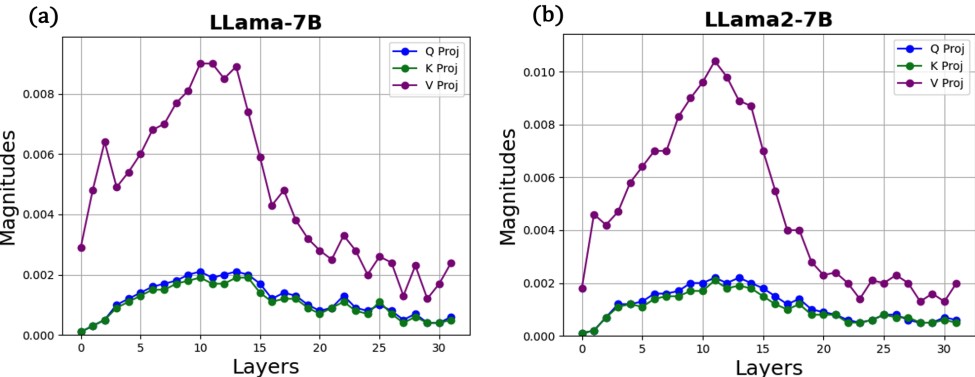

Figure 3: The magnitude of the gradient for the Q, K, and V vectors at each layer in LLMs.

Based on our observation that fine-tuning attention is more efficient during fine-tuning, in all of our SMT experiments in Section§ 4, we only allocate SMT trainable parameters to Q, K, V vectors from attention mechanisms. We rank the average absolute gradient values of every single sub-matrix within attention mechanisms and select those with the highest average values until the parameter ratio limit is reached. Counter-intuitively, we observed that the trainable parameters are predominantly assigned to the V vectors. As shown in Fig. 5, 95.17% of the trainable parameters are automatically

assigned to the V vectors by SMT. Fig. 4 indicates that all V vectors have trainable parameters, while 22 out of 32 Q vectors and 21 out of 32 K vectors are completely frozen.

In our ablation experiments, we experimented with assigning all trainable parameters to only K, or only Q, or only V vectors, and fine-tuned 0.86% of the parameters on LLaMA-7B using the `Commonsense` dataset. Table 6 presents four additional experiments where we fine-tuned 0.86% of the parameters of LLaMA-7B using SMT on the `Commonsense` dataset. In the first three rows, all trainable parameters are allocated to the K vectors, Q vectors, and V vector , respectively. In the fourth row, the trainable parameters are assigned to Q, K, V vectors directly and allocated by SMT automatically. The trainable parameters are distributed among the K, Q, and V vectors, as detailed in Fig. 5, with the trainable states of the QKV layers shown in Fig. 4.

Figure 4: A visualization of trainable Q, K, V layers when fine-tuning 0.86% trainable parameters on LLaMA-7B. LLaMA-7B has 32 layers of MLPs, each contains a Q vector, a K vector, and a V vector. White layers are frozen and green layers contain trainable parameters.

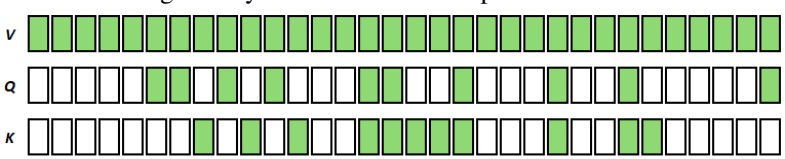

Figure 5: Distribution of trainable parameters among Q, K, V.

Table 6: K SMT, Q SMT, and V SMT assign all trainable parameters to only K, or only Q, or only V vectors respectively, and fine-tuned 0.84% of the parameters on LLaMA-7B using the Commonsense dataset. QKV SMT assign all trainable parameters to QKV vectors and select sub-matrices automatically.

| Model | Param location | #Params% | BoolQ | PIQA | SIQA | HellaSwag | WinoGrande | ARC-e | ARC-c | OBQA | AVG |
|---|---|---|---|---|---|---|---|---|---|---|---|
| | **K SMT** | 0.84 | 65.5 | 79.1 | 76.2 | 88.3 | 73.2 | 80.3 | 60.8 | 68.0 | 73.9 |
| **LLaMA-7B** | **Q SMT** | 0.84 | 65.7 | 79.3 | 75.5 | 88.2 | 72.5 | 80.1 | 59.6 | 72.5 | 75.3 |
| | **V SMT** | 0.84 | 68.7 | 82.1 | 78.1 | 91.6 | 78.8 | 83.0 | 68.7 | 77.2 | 78.5 |
| | **QKV SMT** | 0.84 | 68.7 | 81.7 | 78.3 | 91.6 | 78.8 | 84.1 | 68.7 | 77.4 | 78.7 |

The results show a significant performance gap when comparing the allocation of all trainable parameters to the V vectors versus the Q and K vectors. Assigning all parameters to the V vectors outperforms the K vectors by 4.6% and the Q vectors by 3.2%. These observations suggest that fine-tuning the V vectors is the most efficient compared to Q and K in this process; it also hints that SMT is able to effectively select sub-matrices containing crucial memory sections.

To provide further insight into the importance of V vectors, we visualized the median magnitude of the absolute gradients for each vector across layers, as shown in Fig. 3. The results, averaged over 1000 iterations, are based on two models: LLaMA-7B and LLaMA2-7B. We observe that ***the gradients of the V vectors are significantly larger than those of the Q and K vectors***, with the V vector gradients being up to 10 times greater in most layers. This larger gradient leads to more substantial updates, making fine-tuning the V vectors more effective.

In Appendix C, we show that the smaller magnitude of gradient for the Q and K vectors is caused by the design of scaling in the attention mechanism, where $\sqrt{d_k}$ is not sufficiently large during pre-training.

## 6    CONCLUSION

Our empirical results suggest that fine-tuning attention layers are more critical than MLPs for downstream performance; V vector is the most influential vector for performance among Q, K, V vectors. Overall, our SMT method is an appealing alternative to LoRA especially for practitioners with limited compute resources, since SMT could achieve better accuracy than other SoTA PEFT methods (LoRA, DoRA and SpIEL) with the same amount of trainable parameter. SMT is also a strong alternative vs. full fine-tuning, since we showed that the gap between SMT and full tuning is very narrow.

# 7 ACKNOWLEDGMENTS

We sincerely thank Sida Wang for her valuable contributions to this project, including developing and finalizing the implementation methodology, generating critical experimental results that strengthened our conference presentation, and supporting the open-source release of our codebase. We also extend our gratitude to Qianou (Christina) Ma, Chenyang Yang, Chen Liu, and Yu (Ivy) Yang for the suggestion in paper writing and their valuable feedback.
This research used the Bridges-2 at Pittsburgh Supercomputing Center(PSC) and Delta advanced computing. Pittsburgh Supercomputing Center is supported by National Science Foundation grants #2138259, #2138286, #2138307, #2137603, and #2138296. The Delta advanced computing is supported by the National Science Foundation (award OAC 2005572) and the State of Illinois. Delta is a joint effort of the University of Illinois Urbana-Champaign and its National Center for Supercomputing Applications.
Though Heather Miller and Juncheng Billy Li are employees of Two Sigma Investments, this work was performed independently from Two Sigma Investments.

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

## A    IMPLEMENTATION DETAILS WITH CODE SNIPPETS

```python
class LinearLayer_MatrixSparsity(torch.nn.Module):
    def __init__(self, weight, bias=None, index_list = []):
        super(LinearLayer_MatrixSparsity, self).__init__()
        self.weight = weight
        # freeze all the weight and not passing full weight into optimizer
        self.weight.requires_grad = False
        self.bias = bias
        self.index_list = index_list

        # maintain a new trainable parameter `selected_weight`
        self.selected_weight = torch.empty(len(index_list) * Block_dimension, Block_dimension)
        self.selected_weight.requires_grad = True

        # project original weight parameters to new trainable parameters `selected_weight`
        for i in range(len(index_list)):
            index = index_list[i]
            self.selected_weight[i * Block_dimension: i * Block_dimension + Block_dimension, :] = \
            self.weight.data[index[0] * Block_dimension: index[0] * Block_dimension + Block_dimension, \
                             index[1] * Block_dimension: index[1] * Block_dimension + Block_dimension]
        self.selected_weight = nn.Parameter(self.selected_weight)

        # apply a specialized sparse linear multiplication function
        self.fn = linearZ.apply
```

Figure 6: Implementation of customized sparse linear layer.

```python
class linearZ(torch.autograd.Function):
    @staticmethod
    def forward(ctx, input, selected_weight, matrix_index_list, weight):
        # maintain partial input in `input_list`
        input_list = []
        for index in matrix_index_list:
            input_list.append(input[:, :, index[1]*Block_dimension: index[1]*Block_dimension+Block_dimension])

        # save the partial input(`input_list`) and sub-matrics index(`matrix_index_list``) for backward propagation
        ctx.list1 = input_list
        ctx.list2 = matrix_index_list

        ctx.save_for_backward(weight)
        output = torch.matmul(input, weight.t())

        return output
```

Figure 7: Implementation of customized forward in specialized sparse linear multiplication function.

```python
class linearZ(torch.autograd.Function):
    @staticmethod
    def backward(ctx, grad_output):
        # fetch weight, partial input(`input_list`), and sub-matrics index(`matrix_index_list``)
        weight,  = ctx.saved_tensors
        input_list = ctx.list1
        matrix_index_list = ctx.list2

        # calculate the partial gradient and maintain it in `grad_weight`
        grad_weight = torch.empty(len(input_list) * Block_dimension, Block_dimension)
        for i in range(len(input_list)):
            index = matrix_index_list[i]
            grad_weight[i * Block_dimension: i * Block_dimension + Block_dimension, :] = torch.sum(torch.matmul(grad_output.permute(0, 2, 1)\
                            [:, index[0] * Block_dimension: index[0] * Block_dimension + Block_dimension, :], input_list[i]), dim=0)
        # calculate dl/dz for backward propagation
        grad_input = torch.matmul(grad_output, weight)

        # return gradient for activation and selected sub-matrices
        return grad_input, grad_weight, None, None
```

Figure 8: Implementation of customized backward in specialized sparse linear multiplication function.

## B  SUB-MATRICES SELECTION

SMT ranks the average absolute gradient values within each sub-matrix and selects those with the highest averages. The rationale behind this selection process is to enable SMT to automatically identify sub-matrices containing memory that is most relevant to downstream tasks. During fine-tuning, the absolute gradient values can indicate the relevance of a block to these tasks hence it requires more tuning. By averaging the absolute gradient values within each sub-matrix, we can determine the importance of the sub-matrix to specific downstream tasks. In Appendix B.1, we further discuss why we calculate the average absolute gradient values, and the differences between Fisher Information (Sung et al., 2021) and SMT.

### B.1  FISHER INFORMATION VS. SMT

***What is True Fisher Mask and why is it crucial?*** In the context of neural networks, Fisher information (Sung et al., 2021) measures how much each parameter $\theta$ affects the model's predictions. It is a crucial concept used to determine the importance of parameters for the task at hand. The paper uses Fisher information to identify which parameters to update during training. The Fisher information matrix $F_\theta$ is introduced as follows:

$$\mathbb{E}_x \left[ D_{KL}(p_\theta(y|x) \,\|\, p_{\theta+\delta}(y|x)) \right] = \delta^\top F_\theta \delta + \mathcal{O}(\delta^3) \tag{4}$$

where $p_\theta(y|x)$ represents the output distribution of the model, given input $x$ and parameter vector $\theta$. It represents the probability of the model predicting class $y$ given input $x$; $\delta$ represents a small perturbation applied to the parameter vector $\theta$. It represents a slight change in the parameter values; $F_\theta$ is the Fisher information matrix, which quantifies how sensitive the model's predictions are to changes in each parameter $\theta$; $D_{KL}(p_\theta(y|x) \,\|\, p_{\theta+\delta}(y|x))$ is The Kullback-Leibler (KL) divergence, which measures how much the model's output distribution changes when the parameters are perturbed by $\delta$; $\mathcal{O}(\delta^3)$ represents higher-order terms, which become negligible as $\delta \to 0$. The Fisher information matrix $F_\theta$ is further defined as:

$$F_\theta = \mathbb{E}_{x \sim p(x)} \left[ \mathbb{E}_{y \sim p_\theta(y|x)} \nabla_\theta \log p_\theta(y|x) \nabla_\theta \log p_\theta(y|x)^\top \right] \tag{5}$$

where $\mathbb{E}_{x \sim p(x)}$ represents The expectation over the input data distribution $p(x)$, which captures how much variation exists across different inputs; $\mathbb{E}_{y \sim p_\theta(y|x)}$ represents the expectation over the output distribution $p_\theta(y|x)$ of the model, given the current parameter values $\theta$ and input $x$; $\nabla_\theta \log p_\theta(y|x)$ represents the gradient of the log-likelihood of the model's output with respect to the parameters $\theta$. This measures how sensitive the model's predictions are to changes in the parameters. $\nabla_\theta \log p_\theta(y|x)^\top$ represents the transpose of the gradient, creating a matrix that measures the correlations between different parameters in terms of their effect on the model's output. This matrix captures how much each parameter influences the model's predictions, and the larger its value, the more important that parameter is for the task.

***What is empirical Fisher Mask and how is it different with true fisher mask?*** (Sung et al., 2021) distinguishes between the true Fisher information and an approximation called the *empirical Fisher*. In the empirical Fisher approximation, instead of sampling from the model's output distribution $p_\theta(y|x)$, the known ground truth labels $y_i$ are used directly. The empirical Fisher approximation is defined as:

$$\hat{F}_\theta = \frac{1}{N} \sum_{i=1}^{N} \left( \nabla_\theta \log p_\theta(y_i|x_i) \right)^2 \tag{6}$$

The empirical Fisher can be more computationally efficient because it avoids sampling from the model's predicted distribution. It is used in (Sung et al., 2021) as a heuristic for parameter importance and is found to perform similarly to the true Fisher in practice. To construct the sparse mask based on the empirical Fisher, the top-$k$ parameters with the highest Fisher information are selected. Specifically, the mask selects parameters $\theta_i$ such that:

$$\theta_i \in \left\{ \theta_j \mid \hat{F}_{\theta_j} \geq \text{sort}(\hat{F}_\theta)_{[k]} \right\} \tag{7}$$

Where $k$ is the number of parameters to be updated based on the desired sparsity level, and $\text{sort}(\hat{F}_\theta)_{[k]}$ represents the $k$-th largest Fisher information value. ***How does Empirical Fisher Mask Calculated?*** Fisher information (Sung et al., 2021) uses the top-k accumulation of squared gradient values to identify important parameters, which is mathematically equivalent to our approach of using the average absolute gradient value in SMT. Fisher information requires a fisher mask, $m_F$, defined as below:

$$m_{F,i} = \begin{cases} 1 & \text{if } F_{\theta_i} \geq \text{sort}(F_\theta)_{[k]} \\ 0 & \text{otherwise} \end{cases}$$

This mask is applied as: $\theta_{\text{masked}} = m_F \odot \theta$.

***What is the differences between Fisher Mask and SMT?*** Fisher information (Sung et al., 2021) uses the top-k accumulation of squared gradient values to identify important parameters, which is mathematically equivalent to our GW-Selection approach of using the average absolute gradient value in SMT as following:

$$\hat{F}_{\text{smt},\hat{\theta}} = \frac{1}{N} \sum_{i=1}^{N} \left| \nabla_{\hat{\theta}} \log p_{\hat{\theta}}(y_i|x_i) \right| \tag{8}$$

where $\hat{F}_{\text{smt},\hat{\theta}}$ is the SMT calculation of gradient information, using the mean absolute gradient value for each sub-matrix parameter $\hat{\theta}$; $N$ is The number of data samples; $x_i$ is The $i$-th input data sample; $y_i$ is The ground-truth label corresponding to the $i$-th input; $\nabla_{\hat{\theta}} \log p_{\hat{\theta}}(y_i|x_i)$ is The gradient of the log-likelihood of the model's output with respect to the selected sub-matrix parameter $\hat{\theta}$ for sample $(x_i, y_i)$. Here, we take the absolute value of this gradient.

Since (Sung et al., 2021) needs to store the fisher mask, $m_F$, for all parameters, the memory costs of Fisher Mask are almost the same as the memory costs of the model parameters because the mask is stored as tensors with the same shape as the model parameters. However, instead of storing mask with large memory costs, SMT deletes mask by projecting trainable parameters to dense tensors. This is crucial since the large model size leads to memory bottleneck for GPUs. In addition, to calculate $\hat{F}_\theta$, Fisher Mask requires full backward propagation and calculate gradient for all parameters $\theta$, while SMT only requires partial backward propagation. SMT only needs to calculate the gradient$\hat{F}_{smt,\hat{\theta}}$ for selected parameters $\hat{\theta}$ within sub-matrix. In every fine-tuning iteration of Fisher Mask (Sung et al., 2021), full backward propagation needs to be done. This will lead to significant computational costs and slow down the fine-tuning. However, by customizing linear layers and linear functions, SMT can do partial backward computation and speedup the whole fine-tuning.

Additionally, in Table 7, we further evaluate four different sub-matrix selection methods using gradient information. Our experimental results show that using the average absolute gradient value is the most effective approach for selective tuning within our SMT framework.

Table 7: SMT experiment results using different sub-matrices evaluation metrics on Common-Sense dataset. This table aims to perform an ablation study to find which sub-matrices evaluation metrics is the best. All experiments fine-tune 0.84% overall parameters in Q, K, V vectors. SMT-1 setting: abs().mean() calculation. SMT-2 setting: mean().abs() calculation. SMT-3 setting: abs().sum(dim=(1, 3)) calculation. SMT-4 setting: sqrt(sum(abs() ** 2, dim=(1, 3))) calculation. Conclusion: mean().abs() is the best strategy.

| Model | BoolQ | PIQA | SIQA | HellaSwag | WinoGrande | ARC-e | ARC-c | OBQA | AVG |
|-------|-------|------|------|-----------|------------|-------|-------|------|------|
| SMT-1 | 66.5 | 79.6 | 73.8 | 86.2 | 72.8 | 80.2 | 61.2 | 70.1 | **73.80** |
| SMT-2 | 68.7 | 81.7 | 78.3 | 91.6 | 78.8 | 84.1 | 68.7 | 77.4 | **78.66** |
| SMT-3 | 67.2 | 81.2 | 75.8 | 86.8 | 76.2 | 80.8 | 62.8 | 70.8 | **75.20** |
| SMT-4 | 68.2 | 81.2 | 75.8 | 87.4 | 76.2 | 81.3 | 63.2 | 70.8 | **75.51** |

### B.2 AW-Selection vs. GW-Selection in SMT

In large language models (LLMs), not all weights contribute equally to the model's performance. A small fraction of the weights, referred to as *salient weights*, are significantly more important. We propose two salient weights selection methods in warm up phase, AW-Selection and GW-Selection, and discuss them in this section.

AW-Selection is inspired by AWQ (Lin et al., 2024) which proposes a method to identify these weights by using the activation distribution of the neural network, instead of just considering the magnitude of the weights. ***How we use AW-Selection to select trainable parameters?*** In the following, we explain how AWQ uses the activation information to select these salient weights.

Let us define the forward pass of a single layer in a neural network. Given input activations $\mathbf{x}$ and weight matrix $\mathbf{W}$, the output $\mathbf{y}$ is computed as:

$$\mathbf{y} = \mathbf{W}\mathbf{x}$$

Each column of the weight matrix $\mathbf{W}$ corresponds to a specific channel or neuron in the network, and each weight channel interacts with the input activations $\mathbf{x}$ to produce the output.

The key insight behind AW-Selection is that ***weight channels associated with larger activation magnitudes*** are more important for the model's overall performance. These channels process more important features of the input, and therefore, their corresponding weights need to be selected as trainable parameters.

The importance of a weight channel $W_i$ can be measured by the magnitude of its corresponding activations. Mathematically, the *activation magnitude* for channel $i$ can be represented as:

$$\text{Activation Magnitude for channel } i = \mathbb{E}\left[\|\mathbf{x}_i\|_2\right]$$

where $\mathbf{x}_i$ is the activation corresponding to weight channel $W_i$, and $\mathbb{E}\left[\|\mathbf{x}_i\|_2\right]$ denotes the expected value of the $L_2$ norm of the activations. Intuitively, channels with larger average activations are more important because they carry more information or contribute more heavily to the output.

To identify the salient weights, the AW-Selection involves the following **Salient Weight Selection Strategy: (1)Measure Activation Magnitude:** For each weight channel $W_i$, the corresponding activation magnitude is measured by calculating the average $L_2$ norm of the activations over a calibration dataset. **(2) Rank Channels by Activation Magnitude:** Once the activation magnitudes are computed for all channels, the channels are ranked based on their activation magnitudes. Channels with higher activation magnitudes are ranked higher. **(3)Select Salient Channels:** Based on the ranking, a small fraction of channels (typically 0.1% to 1%) are selected as *salient channels*. These are the channels that correspond to the most important features in the model. Since AW-Selection only requires activations information, the forward propagation is sufficient to gather activations. Since AW-Selection doesn't require backward propagation and optimizer, it leads to less computational costs and memory costs compare to GW-Selection.

Inspired by empirical Fisher derivation (see Appendix B.1 for more details about Fisher information and how SMT differs with it), we propose GW-selection to identify specific submatrices within the model's weight matrices that show the greatest gradient changes during a warm-up phase at the beginning of fine-tuning. Since GW-Selection requires full backward propagation, it updates the base model during the warm-up phase, and identify the salient sub-matrices according to the average of absolute value of gradients. Mathematical formulations for GW-selection can also be found in Appendix B.1.

Further comparisons between fine-tuning results using Activation-aware parameter selection (AW-Selection) and Gradient-aware parameter selection (GW-Selection) are presented in Table 8. A significant performance gap can be observed between AW-Selection and GW-Selection in sparsity fine-tuning. ***AW-Selection tends to overfit more easily than GW-Selection when fine-tuning the same number of trainable parameters under identical hyper-parameter settings.*** Even after adjusting the hyper-parameters, AW-Selection still performs substantially worse than GW-Selection. In Table 8, we present the best results for AW-Selection.

Table 8: Results on fine-tuning LLaMA-7B using gradient-aware parameters selection (GW-Selection) and activation-aware parameter parameters selection (AW-Selection) on CommonSense dataset.

| Model | PEFT method | #Params% | BoolQ | PIQA | SIQA | HellaSwag | WinoGrande | ARC-e | ARC-c | OBQA | AVG |
|---|---|---|---|---|---|---|---|---|---|---|---|
| LLaMA-7B | GW-Select | 0.84 | 68.7 | 81.7 | 78.3 | 91.6 | 78.8 | 84.1 | 68.7 | 77.4 | 78.7 |
| | AW-Select | 0.84 | 62.0 | 49.5 | 52.9 | 70.8 | 48.6 | 54.8 | 41.2 | 45.7 | 53.2 |
| | GW-Select | 4.91 | 72.0 | 82.9 | 80.7 | 93.3 | 82.4 | 86.1 | 70.6 | 83.0 | 81.4 |
| | AW-Select | 4.91 | 62.5 | 50.4 | 48.7 | 68.8 | 50.2 | 56.1 | 40.8 | 44.8 | 52.8 |

**More details about AWQ (Lin et al., 2024):** Once the salient channels are identified based on activation magnitude, AWQ proposes a strategy to protect these channels by scaling them. The following equation (Equation 5 in the AWQ paper) describes how the scaling is performed:

$$s = s_X^\alpha, \quad \alpha^* = \arg\min_\alpha L(s_X^\alpha)$$

where $s_X$ is the average magnitude of the activation (per-channel), and $\alpha$ is a hyperparameter that controls the balance between protecting the salient and non-salient channels. The objective function $L(s_X^\alpha)$ minimizes the output difference after applying the scaling, ensuring that the performance degradation due to weight adjustment is minimized.

The hyperparameter $\alpha$ controls how aggressively the salient channels are protected. When $\alpha$ is larger, the scaling factor $s_X^\alpha$ is more aggressive, placing more emphasis on the channels with larger activations. Conversely, a smaller $\alpha$ results in more balanced protection between salient and non-salient channels. Similarly, in AW-Selection, $\alpha$ is used to control the percentage of trainable parameters. When $\alpha$ is larger, larger percentage of trainable parameters is selected. Conversely, a smaller $\alpha$ results in smaller number of trainable parameters.

### B.3 Ablation Studies on Sub-matrices Selection

In table 9, we present experimental results comparing the following selection methods: selects the top Y% of sub-matrices parameters across all Q, K, V layers; selects the top Y% of sub-matrices parameters within each Q, K, V layer; and random selection, which selects sub-matrices parameters randomly within all Q, K, V layers. Our findings show that automatic top Y% parameter selection across all K, Q, V parameters consistently outperforms the top Y% selection within each Q, K, V layer and the random selection method.

Table 9: Fine-tuning LLama-7B using SMT on CommonSense dataset. This table aims to perform an ablation study to find which parameter selection approach is the best. SMT-1: Select the top 0.84% of parameters from all Q, K, V parameters. SMT-2: Select the top 0.84% of parameters in every layers of Q, K, V vectors. SMT-2: Randomly Select 0.84% of parameters in every layers of Q, K, V vectors.

| Model | BoolQ | PIQA | SIQA | HellaSwag | WinoGrande | ARC-e | ARC-c | OBQA | AVG |
|---|---|---|---|---|---|---|---|---|---|
| SMT-1 | 68.7 | 81.7 | 78.3 | 91.6 | 78.8 | 84.1 | 68.7 | 77.4 | **78.66** |
| SMT-2 | 66.5 | 80.8 | 73.2 | 85.1 | 71.4 | 79.7 | 58.8 | 67.2 | **72.83** |
| SMT-3 | 61.7 | 74.8 | 70.1 | 82.4 | 68.4 | 75.2 | 54.5 | 64.5 | **68.95** |

## C  SMALL GRADIENT CAUSED BY SOFTMAX SATURATES

### C.1  WHAT CAUSE THE SMALL GRADIENT IN Q/ K VECTORS?

In the context of the scaled dot-product attention mechanism, the softmax function is typically applied to the scaled dot-product between the query and key matrices, expressed as:

$$\text{softmax}\left(\frac{QK^\top}{\sqrt{d_k}}\right),$$

where $Q \in \mathbb{R}^{n \times d_k}$ is the query matrix, $K \in \mathbb{R}^{n \times d_k}$ is the key matrix, and $d_k$ is the dimension of the key vectors. The term $QK^\top \in \mathbb{R}^{n \times n}$ represents the matrix of dot products between query and key vectors. The softmax function is applied along the rows of the scaled matrix:

$$S = \frac{QK^\top}{\sqrt{d_k}},$$

where the elements of $S$ are given by

$$S_{ij} = \frac{Q_i \cdot K_j}{\sqrt{d_k}},$$

with $Q_i \cdot K_j$ denoting the dot product between the $i$-th query vector and the $j$-th key vector.

To understand *why the gradient of the softmax function with respect to Q and K vectors becomes small* when $QK^\top$ is large and $\sqrt{d_k}$ is not sufficiently large, we begin by considering the gradient of the softmax function itself. For a given row $S_i$ of the matrix, the softmax function is defined as:

$$\sigma(S_i) = \frac{e^{S_i}}{\sum_k e^{S_k}}.$$

The gradient of $\sigma(S_i)$ with respect to $S_k$ is given by:

$$\frac{\partial \sigma(S_i)}{\partial S_k} = \begin{cases} \sigma(S_i)(1 - \sigma(S_i)) & \text{if } i = k, \\ -\sigma(S_i)\sigma(S_k) & \text{if } i \neq k. \end{cases}$$

Now, to compute the gradient with respect to $Q$, we must apply the chain rule. The gradient of the scaled dot-product $S_{ij} = \frac{Q_i \cdot K_j}{\sqrt{d_k}}$ with respect to $Q_i$ is:

$$\frac{\partial S_{ij}}{\partial Q_i} = \frac{K_j}{\sqrt{d_k}}.$$

Thus, the gradient of the softmax output with respect to $Q_i$ becomes:

$$\frac{\partial \sigma(S_i)}{\partial Q_i} = \sum_j \frac{\partial \sigma(S_i)}{\partial S_{ij}} \cdot \frac{\partial S_{ij}}{\partial Q_i}.$$

Substituting the gradients, we obtain:

$$\frac{\partial \sigma(S_i)}{\partial Q_i} = \sum_j \sigma(S_i) \left(\delta_{ij} - \sigma(S_j)\right) \cdot \frac{K_j}{\sqrt{d_k}}.$$

We now examine the behavior of the gradient when $QK^\top$ becomes large and $\sqrt{d_k}$ is not large enough. If the values of $QK^\top$ are large, the elements $S_{ij}$ will also be large. As $S_{ij}$ grows, the

softmax function saturates, meaning that one of the softmax outputs will approach 1, while the remaining outputs approach 0. In such cases, the gradient of the softmax function becomes very small because for large $S_i$, the output $\sigma(S_i)$ is close to 1 for the largest element and close to 0 for all other elements, leading to the following observations:

1. For the largest $S_i$, the softmax output approaches 1, and the gradient becomes:

$$\frac{\partial \sigma(S_i)}{\partial S_i} = \sigma(S_i)(1 - \sigma(S_i)) \approx 1 \cdot (1 - 1) = 0.$$

2. For the smaller $S_j$ (where $j \neq i$), the softmax output approaches 0, and the gradient also becomes very small:

$$\frac{\partial \sigma(S_j)}{\partial S_j} = \sigma(S_j) \cdot \sigma(S_i) \approx 0.$$

***The small gradient arises because the softmax function compresses a wide range of input values into a narrow range of output probabilities between 0 and 1. As a result, when one value dominates the others, the gradients with respect to the input values shrink significantly.***

Next, consider the impact of the scaling factor $\sqrt{d_k}$. This scaling is introduced to mitigate the effect of large dot products by dividing the dot products $Q_i \cdot K_j$ by $\sqrt{d_k}$. However, if $\sqrt{d_k}$ is not large enough, the elements of $S = \frac{QK^\top}{\sqrt{d_k}}$ can still become large, leading to saturation in the softmax function. As a result, the gradients with respect to $Q$ remain small.

In summary, ***the gradient of the softmax with respect to $Q$ becomes small when $QK^\top$ is large and $\sqrt{d_k}$ is not large enough due to the saturation of the softmax function.*** When the scaled dot-product $\frac{QK^\top}{\sqrt{d_k}}$ has large values, the softmax output tends to concentrate around 1 for one element and near 0 for the others, resulting in near-zero gradients. The scaling factor $\sqrt{d_k}$ alleviates this issue to some extent, but if $d_k$ is small, the problem of small gradients persists, hindering the optimization process.

## C.2   Scaling $\sqrt{d_k}$

We now demonstrate ***how increasing $\sqrt{d_k}$ affects the variance of $\frac{QK^\top}{\sqrt{d_k}}$***, and how an increase in variance leads to larger differences between the elements of $QK^\top$.

Consider the dot-product between query and key vectors $Q_i \cdot K_j$. Let the elements of $Q$ and $K$ be independent and identically distributed (i.i.d.) random variables with mean 0 and variance $\sigma^2$. The dot product $Q_i \cdot K_j$ is the sum of $d_k$ independent terms, each of variance $\sigma^2$. Therefore, by the properties of variance, the variance of $Q_i \cdot K_j$ is:

$$\text{Var}(Q_i \cdot K_j) = d_k \cdot \sigma^2. \tag{9}$$

Now, when we apply the scaling factor $\frac{1}{\sqrt{d_k}}$, the variance of the scaled dot-product becomes:

$$\text{Var}\left(\frac{Q_i \cdot K_j}{\sqrt{d_k}}\right) = \frac{1}{d_k} \cdot \text{Var}(Q_i \cdot K_j) = \frac{d_k \cdot \sigma^2}{d_k} = \sigma^2.$$

Thus, the variance of $\frac{Q_i \cdot K_j}{\sqrt{d_k}}$ remains constant and independent of $d_k$, which stabilizes the variance of the scaled dot-products. However, if $\sqrt{d_k}$ is too small, the variance of the dot-products $QK^\top$ will increase. This increase in variance implies that the differences between the elements of $QK^\top$ also become larger, as larger variance leads to more spread-out values, which accentuates the differences between the individual elements of $QK^\top$. Consequently, if the differences between the elements of $QK^\top$ increase, the softmax function is more likely to saturate, resulting in one element becoming significantly larger than the others, and thus leading to smaller gradients, as discussed earlier.

# D    MATH10K DATASET

`Math10K` dataset can evaluate the effectiveness of LLMs on the arithmetic reasoning task. `Math10K` incorporate six subsets including `GSM8k`, `SingleEq`, `SVAMP`, `MultiArith`, `AddSub`, and `AQuA`.(1) the `GSM8K` (Cobbe et al., 2021) dataset consists of high quality linguistically diverse grade school math word problems created by human problem writers, (2) the `SVAMP` (Patel et al., 2021) benchmark consists of one-unknown arithmetic word problems for up-to-4 grade level students by making simple changes to a set of problems from another existing dataset, (3) the `MultiArith` (Roy & Roth, 2016) dataset of math word problems requiring multiple reasoning steps and operations, (4) the `AddSub` (Hosseini et al., 2014) dataset of addition and subtraction arithmetic word problems, (5) the `AQuA` (Ling et al., 2017) dataset of algebraic word problems with natural language rationales, and (6) the `SingleEq` (Koncel-Kedziorski et al., 2015) dataset of grade-school algebra word problems that map to single equations with varying length;

# E    PLATEAU ISSUE FOR LORA AND DORA

In Fig. 9, we visualize the performance of LoRA, DoRA, and SMT under different scaling of trainable parameters on `CommonSense` dataset.

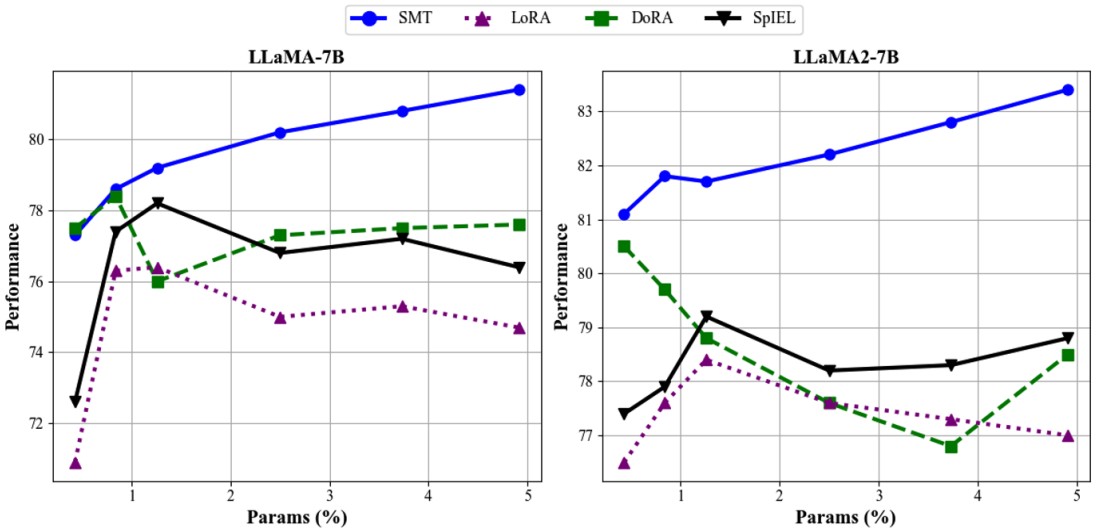

Figure 9: Accuracy comparison of LoRA, DoRA, and SMT under different scaling of trainable parameters on `commonsense` reasoning datasets.

# F    MORE DETAILS ABOUT LORA

In Equation. 1, for a pre-trained weight matrix $W_0$, LoRA constrains its update by representing the latter with a low-rank decomposition $W_0 + \Delta W = W_0 + BA$, where $B \in R^{d \times r}$, $A \in R^{d \times r}$, and the rank $r \ll min(d, k)$.

# G   DENSE MATRIX MULTIPLICATION IN SMT OUTPERFORMS SPARSE MATRIX MULTIPLICATION

In this section, we explain in detail why SMT's use of dense matrix multiplication with concatenated sub-matrices outperforms sparse matrix multiplication, particularly when implemented with PyTorch's `tensor.sparse` format.

## G.1   MEMORY OVERHEAD IN SPARSE MATRICES

Sparse matrices often incur significant memory overhead due to the storage of additional indices, which is not required for dense matrices. Let us consider a concrete example of the LLaMA2 model, where the projection matrices $Q$, $K$, and $V$ have dimensions $4096 \times 4096$.

In the case of a dense matrix, all elements are stored directly, resulting in the need to store $4096 \times 4096 = 16,777,216$ values. However, in a sparse matrix format, if only $5\%$ of the elements are non-zero, then only $0.05 \times 16,777,216 = 838,861$ values need to be stored. Despite this reduction in stored values, sparse matrices require storing the *indices* of non-zero elements, which introduces significant overhead. Specifically, for each non-zero element, two indices (row and column) must be stored, leading to an additional $2 \times 838,861 = 1,677,722$ indices, effectively resulting in sparse matrices taking up to **5 times more memory** than a concatenated dense matrix used in SMT. We illustrate this in Fig. 10 and Fig. 11. In Fig. 10 we present the code we use to test memory and Fig. 11 displays the memory costs for each generated tensor.

The sparse matrix is `Tensor3` in Fig. 11 which contains 12.80M to store $1,677,722$ indices and 3.20M to store $838,861$ elements, totaling 16.0 MB. Instead, the dense matrix `Tensor1` requires only 3.20 MB to store the same $838,861$ elements. This dense storage reduces memory usage by a factor of 5 (16.0 MB / 3.2 MB). `Tensor1` is the full dense matrix with size $4096 \times 4096$.

```python
def memoryTest():
    # random_full_tensor is the dense tensor with size 4096*4096
    # sparse_tensor is the sparse tensor with size 4096*4096
    # dense_tensor is the dense tensor with size 397*2113

    random_full_tensor = torch.randn(4096, 4096).to(DEVICE)
    sparse_tensor = torch.zeros(4096, 4096)
    dense_tensor = torch.randn(397, 2113)
    sparse_tensor[0:397, 0:2113] = dense_tensor
    sparse_tensor = sparse_tensor.to_sparse().to(DEVICE)

    torch.cuda.empty_cache()
    reporter = MemReporter()
    reporter.report()
```

Figure 10: Implementation of memory costs test.

```
Element type                                                Size   Used MEM
---------------------------------------------------------------------------

Tensor0                                               (4096, 4096)    64.00M
Tensor1                                                (397, 2113)     3.20M
Tensor3                                              (2, 838861)      12.80M
Tensor3                                                 (838861,)      3.20M
---------------------------------------------------------------------------
Total Tensors: 20132660     Used Memory: 83.20M
---------------------------------------------------------------------------
```

Figure 11: Memory costs.

By concatenating the selected sparse sub-matrices into a single dense matrix, SMT avoids the need for index storage, thereby achieving significantly lower memory consumption, especially in GPU-constrained environments.

### G.2 NONCONTINUOUS MEMORY IN SPARSE MATRICES

GPUs are optimized for dense matrix operations, benefiting from contiguous memory layout and better parallelization. In dense matrix operations, the data is stored in a continuous block of memory, ensuring that all elements are accessed in a regular and predictable manner. This leads to fewer cache misses and higher throughput during computation.

In contrast, sparse matrices inherently involve irregular memory access patterns because accessing each non-zero element requires retrieving its associated indices. Unlike SMT, which requires only a single index for a submatrix block and benefits from continuous memory within that block, sparse matrices must repeatedly resolve indices for individual elements and map them into contiguous memory. This leads to inefficient memory utilization, lower cache efficiency, and increased latency.

Additionally, sparse matrices often experience load imbalances during parallel computations, as different rows or columns can contain vastly differing numbers of non-zero elements. This imbalance complicates workload distribution across GPU threads, reducing overall parallel efficiency. SMT overcomes these issues by leveraging its structured and dense submatrix memory representation, which minimizes irregular access and ensures higher computational performance. SMT's approach of concatenating sparse sub-matrices into dense matrices enables the use of highly optimized dense matrix multiplication libraries (such as cuBLAS on NVIDIA GPUs), ensuring maximum hardware utilization and significantly faster computations.

### G.3 LACK OF HARDWARE SUPPORT

Modern hardware, especially GPUs, is optimized for dense linear algebra operations using libraries like cuBLAS (NVIDIA) or MKL (Intel). These libraries implement hardware-specific optimizations, such as fused multiply-add (FMA) and matrix blocking, to ensure high performance for dense matrix multiplications.

In contrast, sparse matrix operations, though supported by libraries like cuSPARSE or MKL SPARSE, suffer from hardware inefficiencies due to the irregular memory access patterns inherent in sparse matrices. These irregularities make it challenging to optimize sparse operations to the same degree as dense matrix operations, resulting in lower overall performance.

### G.4 ALGORITHMIC COMPLEXITY

Sparse matrix multiplication algorithms must handle conditional checks to skip zero elements during multiplication, adding overhead to the computation. In dense matrix multiplication, all elements are processed in a straightforward loop without such checks, simplifying the computation.

When sparsity is moderate, the overhead from checking and skipping zero elements in sparse matrices can outweigh the performance gains, making sparse matrix operations less efficient than dense matrix operations, where every element is multiplied and summed without conditions.

By avoiding the overheads of sparse matrix operations and leveraging the efficiency of dense matrix multiplication, SMT achieves faster fine-tuning with lower memory usage.

# H WARM UP ITERATIONS

The number of warm-up iterations is determined by running SMT on the Commonsense dataset for 30, 70, 100, 130, and 160 warm-up iterations. We look at the average of the performance across all tasks, and choose the number of warm-up iterations that has the best overall overage performance, as shown in the table 10 below. We conducted similar experiments on the Math-10K dataset, and found using the same procedure that 30 warm-up iterations leads to the best performance on the Math-10k dataset of tasks. Notably, although we conducted an ablation study to select the number of iterations for different datasets, Table 10 shows that performance variance due to the number of warm-up iterations is minimal. This trend is also observed in the Math-10K dataset in Table 11, suggesting that 100 warm-up iterations can generally be applied across most datasets without additional tuning.

Table 10: SMT experiment results using different warm-up iterations on CommonSense dataset. This table aims to find the best gradient warm-up steps. All experiments are using SMT to fine-tune 3.5% trainable parameters in Q, K, V vectors of LLaMA-7B. SMT-30S: 30 gradient warm-up steps; SMT-70S: 70 gradient warm-up steps. SMT-100S: 100 gradient warm-up steps. SMT-130S setting: 130 gradient warm-up steps. SMT-160S: 160 gradient warm-up steps.

| Model | BoolQ | PIQA | SIQA | HellaSwag | WinoGrande | ARC-e | ARC-c | OBQA | AVG |
|---|---|---|---|---|---|---|---|---|---|
| **SMT-30S** | 69.6 | 83.1 | 79.2 | 92.7 | 81.5 | 84.9 | 69.0 | 78.6 | **79.83** |
| **SMT-70S** | 70.1 | 83.1 | 79.4 | 92.7 | 81.1 | 85.4 | 69.3 | 80.2 | **80.16** |
| **SMT-100S** | 71.0 | 82.3 | 80.0 | 92.9 | 81.8 | 86.2 | 70.4 | 80.2 | **80.60** |
| **SMT-130S** | 69.5 | 82.6 | 79.7 | 92.7 | 81.7 | 85.9 | 69.7 | 81.6 | **80.43** |
| **SMT-160S** | 68.9 | 82.4 | 79.7 | 93.0 | 82.4 | 85.4 | 70.2 | 82.0 | **80.50** |

Table 11: SMT experiment results using different warm-up iterations on Math-10K dataset. This table aims to find the best gradient warm-up steps. All experiments are using SMT to fine-tune 0.86% trainable parameters in Q, K, V vectors of LLaMA-7B. SMT-20S: 20 gradient warm-up steps; SMT-30S: 30 gradient warm-up steps. SMT-50S: 50 gradient warm-up steps. SMT-70S setting: 70 gradient warm-up steps. SMT-100S: 100 gradient warm-up steps.

| Model | PEFT method | #Params% | GSM8k | SingleEq | SVAMP | MultiArith | AddSub | AQuA | AVG |
|---|---|---|---|---|---|---|---|---|---|
| | **SMT-20S** | 0.86 | 34.0 | 84.2 | 52.8 | 91.8 | 84.8 | 23.2 | 61.8 |
| | **SMT-30S** | 0.86 | 34.2 | 84.6 | 53.6 | 91.5 | 85.8 | 23.6 | 62.2 |
| **LLaMA-7B** | **SMT-50S** | 0.86 | 34.4 | 83.7 | 52.8 | 91.0 | 85.2 | 22.6 | 61.6 |
| | **SMT-70S** | 0.86 | 33.8 | 83.8 | 52.8 | 91.3 | 84.4 | 22.8 | 61.5 |
| | **SMT-100S** | 0.86 | 33.6 | 83.9 | 53.1 | 91.3 | 85.4 | 22.8 | 61.7 |

# I  SELECTION FOR SUBMATRIX SIZE

The rational behind 256 x 256 as sub-matrix block size is also under the consideration for generalization and trade off between performance and time costs for non-continuous memory projection.

The choice of a $256 \times 256$ sub-matrix block size is based on generalization and a trade-off between performance and the time costs of non-continuous memory projection.

From a generalization perspective, we aim to select a block size that is applicable across all layers in large language models (LLMs). Specifically, in the LLaMA family, the K, Q, V, and O vectors are designed with sizes such as $4096 \times 4096$ in LLaMA-7B, LLaMA-2-7B, $4096 \times 1024$ for K and V in LLaMA-3-8B, and $5120 \times 5120$ in LLaMA-13B/LLaMA-2-13B. The MLPs have sizes like $4096 \times 11008$ in LLaMA-7B/LLaMA-2-7B, $4096 \times 14336$ in LLaMA-3-8B, and $5120 \times 13824$ in LLaMA-13B/LLaMA-2-13B. Among these weight matrices, $256 \times 256$ is the largest common block size, as 256 is the greatest common divisor of 1024, 4096, 5120, 11008, 14336, and 13824. This block size can not only be used in LLaMA models but also in other LLM families such as Mistral and Phi.

While $256 \times 256$ is the largest common factor, alternative block sizes such as $128 \times 128$ and $64 \times 64$ are also potential candidates. We conducted an experiment, as shown in Tab. 12, to evaluate the performance of fine-tuned LLMs with different sub-matrix block sizes. The results indicate that smaller block sizes yield slightly better performance ( 0.5%), but they introduce additional time costs due to the memory projection required by SMT.

For non-continuous memory projection, SMT transfers sub-matrices to dense memory, which incurs time costs. Sparse matrices are often stored in compressed formats (e.g., Compressed Sparse Row, Compressed Sparse Column), where only non-zero elements and their indices are stored Goharian et al. (2003); Ordonez et al. (2016). Smaller block sizes lead to fewer elements per block, increasing the likelihood of data being scattered across different memory locations, which results in more frequent memory look-ups and higher overhead when assembling these blocks into a dense representation. With a block size of $256 \times 256$, the time costs of memory projection are negligible, accounting for less than 3% of the total forward pass time. Specifically, for a $256 \times 256$ block size, the total forward pass time is `4.9e-05` seconds, and the memory projection time is `1.6e-06` seconds. In contrast, for $128 \times 128$ block size and $64 \times 64$ block size, the memory projection time increases to `5.4e-06` seconds (10%), and `1.5e-05` seconds (24%), respectively. While smaller block sizes provide marginal performance improvements (+0.5 point), they significantly increase time costs. Therefore, we chose $256 \times 256$ as the optimal sub-matrix block size. The profiling results were recorded when fine-tuning 0.84% of LLaMA-7B parameters, in a V vector where 2.1% parameters were selected. All time profiling results are averaged over 100 iterations, after 100 warm-up iterations.

Table 12: Experiment results for SMT with different sub-matrix block sizes when fine-tuning 0.84% of LLaMA-7B parameters on the CommonSense dataset. SMT-256: $256 \times 256$ block size; SMT-128: $128 \times 128$ block size; SMT-64: $64 \times 64$ block size. Other experimental settings are described in the experimental section of the main paper.

| Model | BoolQ | PIQA | SIQA | HellaSwag | WinoGrande | ARC-e | ARC-c | OBQA | AVG |
|---|---|---|---|---|---|---|---|---|---|
| **SMT-256** | 68.7 | 81.7 | 78.3 | 91.6 | 78.8 | 84.1 | 68.7 | 77.4 | 78.7 |
| **SMT-128** | 68.9 | 82.3 | 78.8 | 91.8 | 79.2 | 84.5 | 68.8 | 77.8 | 79.0 |
| **SMT-64** | 69.2 | 82.2 | 78.0 | 91.7 | 79.5 | 85.3 | 69.2 | 78.2 | 79.2 |

