# OpenReview forum: "SMT: Fine-Tuning Large Language Models with Sparse Matrices"
_ICLR.cc/2025/Conference — ICLR 2025 Poster_

### Official Review · Reviewer_kjYr · 2024-10-28

**Soundness:** 3
**Presentation:** 4
**Contribution:** 3
**Rating:** 8
**Confidence:** 4

**Summary:**

The PET method proposed in this paper divides parameter matrices into subblocks. Only a subset of these subblocks are changed in fine-tuning, everything else is frozen. The authors demonstrate convincing advantages over LoRA.

**Strengths:**

- Good empirical evaluation across multiple model sizes and datasets
- Insightful ablations in Sec. 5 and the appendix

**Weaknesses:**

- Some practical questions (e.g. block size and mask generalization) are not addressed
- No comparison with other delta tuning / diff pruning approaches and random masking (https://arxiv.org/abs/2405.02596, https://aclanthology.org/2023.emnlp-main.931.pdf)

**Questions:**

- What is the intention behind using 256x256 sub-matrices? What happens in terms of quality and efficiency when you use smaller (down to 1x1) or larger sub-blocks?
- iiuc the warm-up phase still needs the same memory as full fine-tuning. Can you avoid this phase by reusing sub-block selection from a different [multi-task] experiment for a new task?
- The PET Speed-Ups in Table 1 seem fairly large to me.. I would expect 2-4x speed ups by LoRA, not ~14x. Could you cite work that reports LoRA speed-ups over full fine-tuning in a similar ballpark?
- L223: is the gradient step computation cost really z% cheaper? I understand the argument for a single parameter matrix, but don't you need full gradients in layers closer to the output layer to correctly propagate the gradients to this point? Same for the activation costs in the forward pass (L223)
- Why are some of the full fine-tuning results in Table 2 missing?

---

### Official Review · Reviewer_5QaY · 2024-10-30

**Soundness:** 3
**Presentation:** 3
**Contribution:** 2
**Rating:** 6
**Confidence:** 4

**Summary:**

This paper proposes a new parameter-efficient fine-tuning method which only fine-tunes the task-specific sub-matrix for large language models. The method first identifies the salient sub-matrix for each task with gradient-aware selection strategy. Then, only the salient sub-matrix will be optimized during the fine-tuning stage. The experiments demonstrate that the proposed method can achieve comparable or even better performance than full fine-tuning while requiring updating tiny part of parameters. This work also finds that fine-tuning attention layers are more critical than MLPs and the value matrix v is more influential than the query and key matrix Q, K.

**Strengths:**

1. The sparse matrix tuning method seems reasonable for improving downstream task performance.

2. The results are quite positive and the analyses are extensive.

3. The two interesting findings that attention layers and value matrix are more influential may inspire new insights in the field of parameter-efficient fine-tuning.

**Weaknesses:**

1. This SMT method requires warmup stage to identify the salient sub-matrix in LLMs and the warmup iterations rely on the specific downstream tasks and dataset size. It makes the method somewhat complicated to use. The generalization is limited.

2. Compared to LoRA, this SMT method needs to update a subset of parameters of the original LLMs and it will change the original model, so that the original LLMs cannot be reused for downstream tasks.

**Questions:**

Please see the weaknesses.

---

### Official Review · Reviewer_2m9a · 2024-11-02

**Soundness:** 3
**Presentation:** 3
**Contribution:** 2
**Rating:** 6
**Confidence:** 4

**Summary:**

This paper introduces a novel parameter-efficient fine-tuning (PEFT) method, termed Sparse Matrix Tuning (SMT). SMT builds on insights from prior research suggesting that domain-specific knowledge can be localized and sparsely distributed across layers. SMT identifies the most relevant sparse submatrices by analyzing gradients and then selectively fine-tunes only these submatrices. Experimental results demonstrate that SMT not only outperforms previous PEFT methods but also achieves performance levels close to full parameter fine-tuning. Further analysis reveals that SMT effectively mitigates the performance saturation commonly observed with other PEFT approaches.

**Strengths:**

1. SMT inherits the performance benefits of prior sparse tuning methods while addressing their shortcomings in computational and memory efficiency.
2. Analysis reveals that methods like LoRA, DoRA, and SpIEL suffer from performance saturation as tunable parameters scale, whereas SMT maintains consistent performance without this limitation.
3. The gradient-based parameter selection in SMT enables deeper analysis of critical parameter distributions within LLMs. Results indicate that the Attention module, particularly the value (V) vector, requires more intensive fine-tuning in LLMs. This observation provides valuable insights and could potentially inform future studies in parameter-efficient tuning.

**Weaknesses:**

1. **Gap between Claimed and Actual Contribution:** While the authors claim that SMT aims to "close the performance gap between PEFT and full fine-tuning," SMT’s primary focus appears to be on improving the computational and storage efficiency of existing sparse tuning methods rather than introducing fundamentally new fine-tuning strategies. This creates a disconnect between the paper’s stated goal of performance enhancement and its actual contributions, which seem more oriented toward efficiency. To strengthen the paper, it would be beneficial for the authors to align their claims with the actual contributions more closely.
2. **Lack of Theoretical Support:** The claim that SMT can "close the performance gap between PEFT and full fine-tuning" lacks sufficient theoretical justification. The paper primarily discusses SMT’s implementation details and efficiency improvements without explaining why fine-tuning a sparse matrix should lead to performance gains comparable to full fine-tuning. Providing a theoretical explanation or framework to support this belief would significantly strengthen the paper. For instance, the authors could explore how sparse matrix fine-tuning might preserve essential parameter interactions or discuss relevant properties of sparse representations that could theoretically justify this performance potential.
3. **Missing Baselines:** While the paper primarily focuses on enhancing the computational and storage efficiency of sparse tuning methods, the inclusion of additional sparse tuning baselines would provide a more comprehensive evaluation. Comparing SMT with other sparse tuning methods would highlight its specific advantages in efficiency. Additionally, several recent studies, such as LoRA-GA [1] and LoRA-Pro [2], also aim to bridge the gap between PEFT and full fine-tuning. Including these methods in both the discussion and experiments would allow for a more thorough assessment of SMT's performance and effectiveness relative to other state-of-the-art approaches.

[1] Wang S, Yu L, Li J. LoRA-GA: Low-Rank Adaptation with Gradient Approximation[J]. arXiv preprint arXiv:2407.05000, 2024.

[2] Wang Z, Liang J. LoRA-Pro: Are Low-Rank Adapters Properly Optimized?[J]. arXiv preprint arXiv:2407.18242, 2024.

**Questions:**

1. In Table 2, the authors provide full fine-tuning results only for LLaMA-7B and LLaMA2-7B. Adding full fine-tuning results for additional models would help to better contextualize SMT's performance advantage and strengthen the comparison with full fine-tuning methods.
2. It is unclear how the sparsity hyperparameter $z$ is determined in the experiments. While the #Params% results seem largely influenced by LoRA ranks, SMT should, in principle, allow for a more flexible selection of sparsity levels. Providing details on the criteria or method used to select $z$ would clarify this aspect and help illustrate SMT's potential for adaptable parameter tuning.


Typos and writing issues.

1. In line 072, a colon is missing.
2. In lines 146–148, there is an unnecessary space before the reference term.
3. The use of bold and italic formatting is excessive in multiple sentences. This can make the text appear cluttered and distracting. Reducing the use of bold and italics to highlight only the most essential points will make the writing cleaner and enhance readability.

---

> ### Comment · Reviewer_2m9a · 2024-11-24
>
> I thank the authors for their thoughtful response.
>
> Based on the rebuttal and the appendix, it is evident that the method used to select tunable parameters in SMT is from Fisher information [1], with SMT offering an efficient implementation of this approach. This connection diminishes the following contributions highlighted in the paper:
>
> 1. Narrowing the gap between PEFT and full fine-tuning.
> 2. Avoiding the performance saturation observed in LoRA methods.
> 3. Analyzing the distribution of critical parameters within LLMs.
>
> Regarding Contribution 1, which is presented as the primary contribution of the paper, prior work has already demonstrated that Fisher information enables performance comparable to full fine-tuning with only 0.5% of tunable parameters (refer to Table 1 in [1]). Contributions 2 and 3 similarly arise directly from the tunable parameter selection strategy, which is not entirely novel in this context.
>
> While I appreciate the effort involved in developing an efficient implementation and acknowledge the value of this contribution, the limited novelty of the work impacts its overall assessment. Consequently, I will maintain my current score.

---

> ### Comment · Reviewer_2m9a · 2024-11-25
>
> I thank the author for getting back to me so quickly :)
>
> From my point of view, there are two key factors for the proposed method: the parameter selection strategy and the actual implementation.
>
> Regarding the first factor, given that the selection strategy of SMT is mathematically equal to the Fisher information, the performance advantages can hardly be claimed as a unique contribution. While the authors argue that they have also investigated the AW-selection strategy, its performance is evidently low and thus excluded from the current SMT method.
>
> For the second factor, I acknowledge that SMT provides a highly efficient implementation. Since the efficiency issue has been a pain point for sparse fine-tuning methods, this implementation could indeed be helpful for other fields, also exploiting the advantage of sparsity. This is considered a unique contribution.
>
> However, in the abstract and introduction of the current paper, the writing is focused on the performance advantage against other PEFT methods without clearly explaining that this advantage is actually inherited from Fisher information. This could be potentially misleading. When it comes to the method section, the author seems to avoid directly showing their selection strategy and turn to the implementation details. I believe this writing strategy suggests that the author also knows that their core contribution is not in the selection strategy.
>
> I suggest the authors refine their framing and claims and focus more on their unique contributions and findings, especially in the abstract and introduction. Judging from the current paper, I think it falls into borderline interval. I would love to increase my score if the authors could address this issue by submitting a revised paper.

---

> ### Comment · Reviewer_2m9a · 2024-11-29
> **Raised the score**
>
> Considering the updates, I decide to raise my score to 6.

---

### Official Review · Reviewer_ovmN · 2024-11-05

**Soundness:** 3
**Presentation:** 2
**Contribution:** 2
**Rating:** 5
**Confidence:** 3

**Summary:**

This paper introduces Sparse Matrix Tuning (SMT), a parameter-efficient fine-tuning (PEFT) method for large language models (LLMs), aiming to reduce computational and memory costs while maintaining performance close to full fine-tuning (FT). SMT selectively updates only the most significant submatrices in a model's weight matrix, chosen based on gradient changes, enabling efficient tuning without compromising on model accuracy. The authors compare SMT with established PEFT methods such as LoRA and DoRA, demonstrating that SMT outperforms these methods in both accuracy and computational efficiency, especially by overcoming the performance plateau that other low-rank adaptation methods encounter with increased parameters.

**Strengths:**

1. This work contributes to the growing field of parameter-efficient fine-tuning, a critical area for the advancement of LLMs. Efficient tuning techniques like SMT can play a major role in making large models more accessible and practical for diverse applications.

2. The paper provides an insightful analysis of activation-awareness and gradient-awareness methods, ultimately selecting the gradient-based approach as more effective for identifying meaningful submatrices. This comparison enriches the understanding of PEFT techniques and optimizations.

3. The SMT approach offers an improved strategy for identifying impactful sub-networks, which potentially boosts fine-tuning efficacy by focusing on parameters that most influence downstream performance.

**Weaknesses:**

1. While SMT improves on existing PEFT methods, the approach of selecting sub-networks for fine-tuning is not particularly novel, as similar sparse selection methods have been explored previously in transfer learning and model pruning.

2. The paper does not address whether SMT offers better resistance to catastrophic forgetting compared to other PEFT methods. This omission leaves questions about SMT's stability and robustness in long-term, iterative fine-tuning scenarios.

3. The paper lacks some necessary conceptual explanations and mathematical formulations, such as detailed descriptions of submatrix selection and significance metrics. This limited detail could hinder reproducibility and a deeper understanding of SMT's implementation and optimizations.

**Questions:**

None

---

### Official Review · Reviewer_TAx9 · 2024-11-05

**Soundness:** 3
**Presentation:** 2
**Contribution:** 3
**Rating:** 6
**Confidence:** 2

**Summary:**

This paper presents a novel PEFT method, Sparse Matrix Tuning. SMT uses a gradient based criteria to identify the most relevant submatrices of the model to update, and updates only those while freezing the remaining params during training. This allows for memory and speed gains during training which are similar or better to those made by other PEFT methods. SMT performs well compared to strong baseline PEFT methods LoRA DoRA and SpIEL.

**Strengths:**

The problem addressed is important, the evaluation results are good. The approach could be significant to the field.

**Weaknesses:**

The method is poorly described in the section dedicated to its description. What exactly is happening during warm-up phase? It seems that the warm up phase involves taking 100 steps (on the target dataset?) *without updating the model params*, and taking the average of abs-value of gradients for each param of the model to identify sub-matrices. This is critical to specifying the methodology, and should be clearly stated, not deduced by the reader.

No experiments show ablations on the specification criteria, or the size of the 256x256 submatrix. The justification for restricting SMT to the KQV of the attention mechanism is relatively weak. The sole justification for restricting the SMT trainable params to the attention is a strange ablation comparing hardcoded percentages interpolating between only MLP and only attention. A more natural ablation would allow the selection criteria to place trainable submatrices wherever in the entire model as a baseline, and compare to a variety of restricted alternatives.

The writing throughout is relatively poor. Articles are frequently missing, pluralization is often wrong, tense is frequently switched. This can easily be fixed, and would greatly improve readability.

Generally, the method is sound, though poorly described. The experiments have good results, but the analysis leaves some to be desired.

**Questions:**

The implementation specifies (L242) "we use 256 x 256 as sub-matri[x] block size". L260 states "Despite employing
matrix sparsity, we still leverage the advantages of dense matrix multiplication." What happens when it is per-parameter (ie 1x1)? Is there a tradeoff between method performance and efficiency being made in selecting sub-matrices instead of free parameters?  It seems the only reason to have blocks is computational simplicity. The motivation behind this choice should be made explicit, and the tradeoff investigated. What happens with a larger block size?

Nits:
Figure 1 contrasts two techniques, but appears on first glance to be 4 boxes equally spaced. Consider modifying the figure to make more visually apparent the separateness of the techniques.
L 054: submatrix -> submatrices, right?
L 147: the whole pre-trained weight -> all the pretrained weights?
L 163: for downstream -> for *improving* downstream
L 238: sum up gradient -> sum up the gradient / sum up gradients
L 239: iterations -> iteration, sum up gradient information are -> summed up gradient information is
L 387:  "commonsense reasoning datasets" in Table 3 vs COMMONSENSE in Table 4... are these the same? If so, refer using same term
Table 3 should be a plot.
L426-429: this is highly repetetive, please revise
Throughout: please stay within a single tense within a single paragraph (L302-317 especially, but also elsewhere)
Throughout: missing articles, incorrect pluralization or lack thereof

---

### Comment · Area_Chair_MwNE · 2024-11-21
**Reminder: Please respond and update the score if necessary**

Dear Reviewers,

Kindly ensure that you respond proactively to the authors' replies so we can foster a productive discussion. If necessary, please update your score accordingly. We greatly appreciate the time and effort you’ve dedicated to the review process, and your contributions are key to making this process run smoothly.

Thank you,

AC

---

### Meta-Review · Area_Chair_MwNE · 2024-12-22

**Metareview:**

This paper introduces Sparse Matrix Tuning (SMT), a parameter-efficient fine-tuning method for large language models. By selectively updating the most significant submatrices based on gradient analysis, SMT reduces computational and memory costs while maintaining performance levels close to full fine-tuning. The method outperforms established PEFT approaches like LoRA and DoRA, effectively mitigating the performance plateau seen in other methods. SMT emphasizes the importance of fine-tuning attention layers, showing that the value matrix is particularly influential. Overall, SMT offers significant efficiency gains without sacrificing accuracy.

On the positive note, this paper advances the field of parameter-efficient fine-tuning (PEFT) for large language models (LLMs) through SMT. SMT uses a gradient-based method to identify and update the most impactful submatrices, enhancing computational and memory efficiency while maintaining performance. It overcomes limitations of previous methods, which suffer from performance saturation as tunable parameters increase. SMT provides new insights, particularly highlighting the importance of fine-tuning attention layers and the value (V) vector, which may inform future research. The method is supported by comprehensive empirical evaluations and extensive analyses, demonstrating its potential to improve downstream task performance effectively. Based on the aforementioned review, I would be leaning to accept the paper.

**Additional Comments On Reviewer Discussion:**

Reviewer TAx9 highlights that the paper addresses an important problem and presents promising results, with potential significance in the field. However, the method description lacks clarity, particularly during the warm-up phase, and there is insufficient ablation on submatrix specification and justification for focusing on the KQV of the attention mechanism. Writing quality is also noted as needing improvement.

Reviewer ovmN acknowledges the paper's contribution to parameter-efficient fine-tuning (PEFT) and its analysis of gradient and activation awareness, but notes the lack of novelty in sub-network selection and insufficient discussion on resistance to catastrophic forgetting and conceptual explanations.

Reviewer 2m9a credits SMT with addressing efficiency limitations of previous methods and providing insights into fine-tuning attention modules. However, they point out a disconnect between claimed performance improvements and actual contributions, with a need for more theoretical justification and inclusion of additional sparse tuning baselines like LoRA-GA and LoRA-Pro.

Reviewer 5QaY finds the results positive and the approach insightful, but notes that SMT's complexity and limited generalization arise from its reliance on task-specific warm-up stages. Additionally, the method's parameter updates alter original LLMs, hindering their reuse for other tasks.

Reviewer kjYr commends empirical evaluations and insightful ablations but mentions that practical concerns (e.g., block size) are not addressed, and comparisons with other delta tuning or diff pruning methods are lacking.

---

### Decision · Program_Chairs · 2025-01-22

Accept (Poster)